# Visualizing the Emergence of Intermediate Visual Patterns in DNNs

**Mingjie Li**
Shanghai Jiao Tong University
`limingjie0608@sjtu.edu.cn`

**Shaobo Wang**
Harbin Institute of Technology
`181110315@stu.hit.edu.cn`

**Quanshi Zhang**[*]
Shanghai Jiao Tong University
`zqs1022@sjtu.edu.cn`

## Abstract

This paper proposes a method to visualize the discrimination power of intermediate-layer visual patterns encoded by a DNN. Specifically, we visualize (1) how the DNN gradually learns regional visual patterns in each intermediate layer during the training process, and (2) the effects of the DNN using non-discriminative patterns in low layers to construct disciminative patterns in middle/high layers through the forward propagation. Based on our visualization method, we can quantify knowledge points (*i.e.* the number of discriminative visual patterns) learned by the DNN to evaluate the representation capacity of the DNN. Furthermore, this method also provides new insights into signal-processing behaviors of existing deep-learning techniques, such as adversarial attacks and knowledge distillation.

## 1 Introduction

Deep neural networks (DNNs) have achieved superior performance in various tasks, but the black-box nature of DNNs makes it difficult for people to understand its internal behavior. Visualization methods are usually considered as the most direct way to understand the DNN. Recently, several attempts have been made to visualize the DNN from different aspects, *e.g.* illustrating the visual appearance that maximizes the prediction score of a given category [50, 66, 35], inverting intermediate-layer features to network inputs [12], extracting receptive fields of neural activations [73], estimating saliency/importance/attribution maps [74, 45, 75, 32], visualizing the sample distribution, such as PCA [38], t-SNE [54], etc.

In spite of above explanations of the DNN, there is still a large gap between visual explanations of the patterns in the DNN and the theoretical analysis of the DNN's discrimination power. In other words, visualization results usually cannot reflect the discrimination power of features in the DNN.

Therefore, instead of simply visualizing the entire sample, we divide intermediate-layer features into feature components, each of which represents a specific image region. We visualize the discrimination power of these feature components, and we consider discriminative feature components as knowledge points learned by the DNN. Based on above methods, we can diagnose the feature representation of a pre-trained DNN from the following perspectives.

• We visualize the emergence of intermediate visual patterns in a temporal-spatial manner and evaluate their discrimination power. (1) We visualize how the discrimination power of each individual visual

---

[*]Corresponding author. This work was done under the supervison of Dr. Quanshi Zhang. He is with the John Hopcroft Center and the MoE Key Lab of Artificial Intelligence, AI Institute, at the Shanghai Jiao Tong University, China.

35th Conference on Neural Information Processing Systems (NeurIPS 2021).

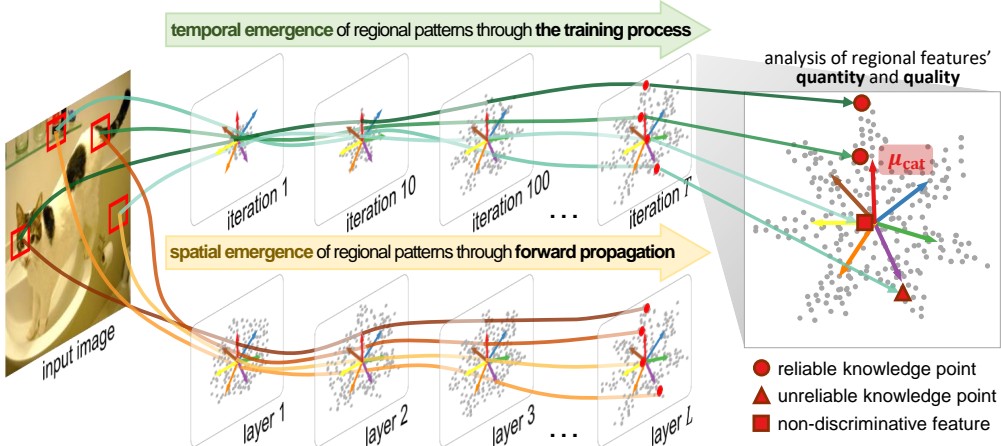

Figure 1: Diagrammatic sketch for the emergence of visual patterns encoded by the DNN in a temporal-spatial manner. The visualization result enables people to analyze the quantity and quality of intermediate features.

pattern increases during the learning process. (2) We illustrate effects of using non-discriminative patterns in low layers to gradually construct discriminative patterns in high layers during the forward propagation. As Figure 1 shows, the regional feature of the cat head emerges as a discriminative pattern for the cat category, while wall features are non-discriminative.

• Based on the the visualization result, we can further measure the quantity and quality of intermediate patterns encoded by a DNN. In Figure 1, we count knowledge points encoded in a DNN as regional patterns with strong discrimination power, and further evaluate whether each knowledge point is reliable for classification. This provides a new perspective to analyze the DNN.

A distinct contribution of this study is to bridge the empirical visualization and the quantitative analysis of a DNN's discrimination power. In comparison, Kim et al. [23] used concept activation vectors to model the relationship between visual features and manually annotated semantic concepts. Cheng et al. [7] quantified the number of visual concepts encoded by the DNN. However, these two methods cannot reflect the discrimination power of regional visual concepts. On the other hand, some researchers derived mathematical bounds on the representational power of a DNN [68, 13] under certain assumptions of the network architecture. To this end, we believe that bridging regional patterns and a DNN's discrimination power is a more convincing and more intuitive way to reveal the internal behavior of a DNN than mathematical bounds under certain assumptions.

Besides, our method provides insightful understanding towards existing deep-learning techniques, such as adversarial attacks and knowledge distillation. (1) For adversarial attacks, we discover that adversarial attacks mainly affect unreliable regional features in high layers of the DNN. The visualization result also enables us to categorize attacking behaviors of all image regions into four types. (2) For knowledge distillation, we discover that the student DNN usually encodes less reliable knowledge points, compared with the teacher DNN. Although knowledge distillation is able to force the student DNN to mimic features of a specific layer in the teacher DNN, there is still a big difference of features in other layers between the student DNN and the teacher DNN.

**Contributions** of this paper can be summarized as follows. (1) We propose a method to visualize the discrimination power of intermediate-layer features in the DNN, and illustrate the emergence of intermediate visual patterns in a temporal-spatial manner. (2) Based on the visualization result, we quantify knowledge points encoded by a DNN. (3) The proposed method also provides new insights into existing deep-learning techniques, such as the adversarial attack and the knowledge distillation.

## 2   Related work

**Visual explanations for DNNs.** Visualization of DNNs is the most direct way to explain visual patterns encoded in the DNN. Some studies reconstructed the network input based on a given intermediate-layer feature [34, 48, 12], while others aimed to generate an input image that cause high activations in a given feature map [67]. Zhou et al. [73] extracted the actual image-resolution receptive

field of neural activations in DNNs. Recently, Goh et al. [17] visualized the multi-modal neurons in the CLIP models [39]. Kim et al. [23] proposed TCAV to represent manually annotated semantic concepts. Another type of researches diagnosed and visualized the pixel-wise saliency/importance/attribution on real input images [42, 32, 24, 75, 14, 74, 5]. What is more, the distribution of intermediate-layer features could be visualized via dimensionality reduction methods, *e.g.* PCA [38], LLE [43], MDS [10], ISOMAP [52], t-SNE [54], etc. Recently, Law et al. [25] visualized the feature of each sample in a low-dimensional space by exploiting probability scores of the DNN, while Li et al. [27] achieved this by assigning each category in the task with a main direction. Sophisticated interfaces have been built up to visualize the architecture and the knowledge of DNNs [19, 60, 53].

Instead of visualizing visual appearance or merely visualizing the distribution of sample features, our method visualizes the emergence of intermediate visual patterns in a temporal-spatial manner, which provides a new perspective to explain DNNs.

**Theoretical analysis of the representation capacity of DNNs.** Formulating and evaluating the discrimination power of DNNs is another direction to explain DNNs. The information-bottleneck theory [64, 47] provides a generic metric to quantify information encoded in DNNs, which is further exploited to evaluate the representation capacity of DNNs [18, 65]. Achille and Soatto [1] further used the information-bottleneck to improve the feature representation. Furthermore, several metrics were also proposed to analyze the robustness or generalization capacity of DNNs, such as the CLEVER score [62], the stiffness [15], the sensitivity metrics [37], etc. Zhang et al. [69] studied the role of different layers towards the generalization of deep models. Based on this, module criticality [4] was proposed to analyze a DNN's generalization power. Cheng et al. [7] quantified visual concepts in input images from the perspective of pixel-wise entropies. Liang et al. [29] diagnosed feature representation between different DNNs via knowledge consistency. Some studies also theoretically proved the generalization bound for two-layer neural networks [68, 13, 36] and deep CNNs [31, 28].

In fact, our research group led by Dr. Quanshi Zhang have proposed game-theoretic interactions as a new perspective to explain the representation capacity of trained DNNs. The interactions have been used to explain the hierarchical structure of information processing in DNNs [70, 72], generalization power [71], the complexity [8] and the aesthetic level [9] of the encoded visual concepts, adversarial robustness [40, 58], and adversarial transferability [57] of DNNs. The interaction was also used to learn baseline values of Shapley values [41].

However, there is still a lack of connections between the explanations of visual concepts and the analysis of a DNN's discrimination power. To this end, our method enables people to use the discrimination power of local regions to explain the overall discrimination power of the entire DNN.

## 3 Algorithm

Given a pre-trained DNN, we propose an algorithm to project the feature of an entire sample and features corresponding to different image regions into a low-dimensional space, in order to visualize (1) how the discrimination power of a sample feature increases during the learning process, (2) effects of the DNN constructing discriminative features in high layers using non-discriminative features in low layers, and (3) how discriminative regional features gradually emerge during the training process. The visualization result reflects whether the sample feature and each regional feature are biased to incorrect categories, which regional features play a crucial role in classification, whether a regional feature is shared by multiple categories, etc. Furthermore, the visualization enables people to quantify knowledge points encoded in the DNN. Such analysis can help people explain existing deep-learning techniques, *e.g.* adversarial attacks and knowledge distillation.

### 3.1 Radial distribution to analyze the discrimination power of features

**Preliminaries: the vMF distribution.** Given a pre-trained DNN and an input image $x \in \mathbb{R}^n$, let us consider the output feature of a specific layer, denoted by $f \in \mathbb{R}^d$. To analyze the discimination power of $f$, previous literature usually considered the feature to follow a radial distribution of massive pseudo-categories (much more than the real category number) [61, 56, 25]. As Figure 1(right) shows, each category/pseudo-category $c$ has a mean direction $\mu_c$ ($c = 1, ..., C$) in the feature space. The significance of classifying $x$ towards category $c$ is measured by the projection $f^\top \mu_c$, and $\cos(f, \mu_c)$ indicates the similarity between $f$ and category $c$. For example, a typical case is the softmax operation,

$p(y|f) = \text{Softmax}(Wf)$, $W \in \mathbb{R}^{C \times d}$. The $c$-th row of $W$ indicates the direction corresponding to the $c$-th category.

To this end, the von Mises-Fisher (vMF) mixture model [3, 20] was proposed to model the radial distribution, where the $c$-th mixture component assumes that the likelihood of each feature $f$ belonging to category $c \in \{1, ..., C\}$ follows a vMF distribution.

$$p(f) = \sum_c p(y = c)p_{\text{vMF}}(f|y = c), \quad p_{\text{vMF}}(f|y = c) = C_d(\kappa_c) \cdot \exp[\kappa_c \cdot \cos(\mu_c, f)], \quad (1)$$

where $C_d(\kappa_c) = \frac{\kappa_c^{d/2-1}}{(2\pi)^{d/2}I_{d/2-1}(\kappa_c)}$ is the normalization constant. Actually, the vMF distribution can be considered as a spherical analogue to the Gaussian distribution on the unit sphere. $\mu_c \in \mathbb{R}^d$ measures the mean direction of category $c$. The increase of $\kappa_c \geq 0$ decreases the variance of $f$'s orientation w.r.t. the mean direction $\mu_c$. Please see the supplementary material for more details.

**Radial distribution with noise.** The vMF distribution assumes that $f$ is a clean feature without noise, which makes the inference of $f$ purely based on its orientation and independent of the strength $\|f\|_2$. However, in real applications, $f$ usually contains the meaningful and clean feature $f^\star$ and the meaningless noise $\epsilon \sim \mathcal{N}(0, \sigma^2 I_d)$, i.e. $f = f^\star + \epsilon$. The existence of the noise decreases the classification confidence if the strength of $f$ is low. Therefore, we have

$$p(f|y = c) = \int p(\epsilon) \cdot p_{\text{vMF}}\Big(f^\star = f - \epsilon|y = c\Big) \, d\epsilon. \quad (2)$$

For simplicity, we can assume that all features $f$ of a specific strength $l$ follow a vMF distribution, because they have similar vulnerabilities to noises. Specifically, let $f = [o, l]$, where $l = \|f\|_2$ and $o = f/l$ represent the strength and orientation of $f$. Then, the likelihood of $f$ belonging to category $c$ is given as follows (proof in the supplementary material).

$$p(f = [o, l]|y = c) = p(l|y = c) \cdot p_{\text{vMF}}(o|y = c, l) = p(l|y = c) \cdot p_{\text{vMF}}(o|\mu_c, \kappa(l)), \quad (3)$$

where $\kappa(l)$ increases along with $l = \|f\|_2$, and $p(l|y = c)$ is the prior distribution of $\|f\|_2$ for category $c$. The variance parameter $\kappa(l)$ is determined based on statistics of all features of the same strength $l$. We can prove the classification result $p(y = c|f)$ is confident when the feature $f$ has a large strength.

## 3.2 Visualization of the sample-wise discrimination power

In this section, we visualize the discrimination power of the feature $f \in \mathbb{R}^d$ of each entire sample $x \in X$. The visualization is supposed to illustrate the classification confidence of each feature towards different categories. Therefore, the goal is to learn a linear transformation to project $f$ into a low-dimensional space, i.e. $g = Mf \in \mathbb{R}^{d'}$ ($d' \ll d$), which ensures that the similarity between each sample feature $f$ and different categories is preserved. The basic idea of learning the linear transformation $M$ is to use the projected feature $g$ for classification, and to force the classification based on $g$ to mimic the classification based on the original feature $f$. Let $y \in Y = \{1, ..., C\}$ denote a category. Thus, the objective is to minimize the KL divergence between the classification probability of the DNN $P(y|x)$ and the classification probability $Q_M(y|x)$ based on the projected feature $g$.

$$\min_M KL[P(Y|X)\|Q_M(Y|X)] \Rightarrow \min_M \mathbb{E}_x\left[\sum_y P(y|x)\log\frac{P(y|x)}{Q_M(y|x)}\right], \quad (4)$$

where $P(y|x)$ is usually computed using a softmax operation. $Q_M(y|x)$ is computed by assuming the distribution $p(g)$ as a mixture model, where each mixture component $p(g|y)$ follows a revised vMF distribution in Eq. (3). Let $g = [l_g, o_g]$, where $l_g = \|g\|_2$ and $o_g = g/l_g$ denote the strength and orientation of $g$. Then, we have

$$p(g) = \sum_y \pi_y \cdot p(l_g|y) \cdot p_{\text{vMF}}(o_g|\mu_y, \kappa(l_g)), \quad (5)$$

where $\pi_y$ denotes the prior of the $y$-th category. We assume the prior of $g$'s strength is independent with the category, i.e. $p(l_g|y) = p(l_g)$. Then, $Q_M(y|x)$ can be measured by the posterior probability $p(y|g)$ in the mixture model, i.e.

$$Q_M(y|x) = \frac{p(y) \cdot p(g|y)}{p(g)} = \frac{\pi_y \cdot p_{\text{vMF}}(o_g|\mu_y, \kappa(l_g))}{\sum_{y'} \pi_{y'} \cdot p_{\text{vMF}}(o_g|\mu_{y'}, \kappa(l_g))}. \quad (6)$$

The training of the sample-level visualization alternates between the following two steps. (i) Given the current linear transformation $M$, we update the mixture-model parameters $\{\pi, \mu\} = \{\pi_y, \mu_y\}_{y \in Y}$ via the maximum likelihood estimation (MLE) $\max_{\{\pi, \mu\}} \prod_g p(g)$. (ii) Given the current state of $\{\pi, \mu\}$, we update $M$ to minimize the KL divergence $KL(P(Y|X)\|Q_M(Y|X))$ in Eq. (4). The supplementary material provides more discussions and derivations about the learning process.

## 3.3 Visualization of the regional discrimination power

In this section, we visualize the discrimination power of features extracted from different regions in each input sample. Let $f \in \mathbb{R}^{K \times H \times W}$ be an intermediate-layer feature of a sample $x$, which is composed of $HW$ regional features for $HW$ positions. Each $r$-th regional feature is a $K$-dimensional vector, and is supposed to mainly describe the $r$-th region in the input, corresponding to the receptive field (region) of $f^{(r)}$. In fact, the actual receptive field of $f^{(r)}$ is much smaller than the theoretical receptive field [73]. The discrimination power of each regional feature $f^{(r)}$ is analyzed in terms of both the *importance* and *reliability*, *i.e.* (1) whether $f^{(r)}$ has a significant impact on the classification, and (2) whether $f^{(r)}$ pushes the classification towards the ground-truth category without significant bias.

The visualization of regional discrimination power needs two overcome to challenges. First, we need to formulate and estimate specific importance of different regions for inference during the learning of visualization. Second, the visualization of regional discrimination power is supposed to be aligned with the coordinate system for sample-wise discrimination power.

The goal of the visualization is to project regional features into a low-dimensional space via a linear transformation $\Lambda$, *i.e.* $h^{(r)} = \Lambda f^{(r)} \in \mathbb{R}^{d'}$ $(d' \ll K)$. Each projected regional feature $h^{(r)}$ is supposed to reflect the *importance* and *reliability* of the original feature $f^{(r)}$. The strength $\|h^{(r)}\|_2$ reflects the importance, and the orientation of $h^{(r)}$ represents the reliability of classification towards different categories. Just like t-SNE [54], we use the projected features $\boldsymbol{h} = \{h^{(1)}, ..., h^{(HW)}\}$ to infer the similarity between samples. In this study, the distinct idea of learning $\Lambda$ is to use the *regional similarities* (based on $\boldsymbol{h}$) as the hidden mechanism of mimicking the *sample-wise similarity*. Let $x_1, x_2 \in X$ be two samples, and the probability of $x_2$ conditioned on $x_1$ represents the sample-wise similarity of $x_2$ to $x_1$. Then, the objective of learning $\Lambda$ is to minimize the KL divergence between the conditional probability $P(x_2|x_1)$ inferred by the DNN and the conditional probability $Q_\Lambda(x_2|x_1)$ inferred by the projected regional features $\boldsymbol{h}$. In this way, each regional feature $h^{(r)}$ can well reflect feature representation $f^{(r)}$ used by the DNN.

$$\mathcal{L}_{\text{similarity}} = KL[P(X_2|X_1)\|Q_\Lambda(X_2|X_1)] \Rightarrow \frac{\partial \mathcal{L}_{\text{similarity}}}{\partial \Lambda} = -\mathbb{E}_{p_{\text{data}}(x_1)}\left[\mathbb{E}_{P(x_2|x=x_1)} \frac{\partial \log Q_\Lambda(x_2|x_1)}{\partial \Lambda}\right] \quad (7)$$

$P(x_2|x_1)$ reflects the similarity of $x_2$ to $x_1$ encoded by the DNN, which is computed using DNN's categorical outputs $z_2, z_1 \in \mathbb{R}^C$. We assume $z_2$ follows a vMF distribution with mean direction $z_1$, *i.e.* $P(x_2|x_1) = \frac{1}{Z}\exp[\kappa_p \cdot \cos(z_2, z_1)]$, where $Z = \sum_{x_2}\exp[\kappa_p \cdot \cos(z_2, z_1)]$.

$Q_\Lambda(x_2|x_1)$ reflects the similarity of $x_2$ to $x_1$ inferred by the projected regional features $\boldsymbol{h}_2$ and $\boldsymbol{h}_1$. Just like the bag-of-words model [51, 11], each projected regional feature $h_2^{(r)}$ is assumed to independently contribute to the inference of $Q_\Lambda(\boldsymbol{h}_2|\boldsymbol{h}_1)$ to simplify the computation. Furthermore, $h_2^{(r)}$ is weighted by its importance $w_2^{(r)} > 0$, *i.e.* $Q_\Lambda(x_2|x_1) \propto \prod_r Q_\Lambda(h_2^{(r)}|\boldsymbol{h}_1)^{w_2^{(r)}}$. Just like in [59], a large value of $w_2^{(r)}$ means the $r$-th region in $x_2$ is important for inference and peaks $h_2^{(r)}$'s contribution $Q_\Lambda(h_2^{(r)}|\boldsymbol{h}_1)$, while a weight $w_2^{(r)}$ near zero flattens out $Q_\Lambda(h_2^{(r)}|\boldsymbol{h}_1)$. Details of the estimation of $w_2^{(r)}$ will be introduced later. In this way, we have

$$\frac{\partial \log Q_\Lambda(x_2|x_1)}{\partial \Lambda} = \sum_r w_2^{(r)} \frac{\partial \log Q_\Lambda(h_2^{(r)}|\boldsymbol{h}_1)}{\partial \Lambda}, \quad (8)$$

where $Q_\Lambda(h_2^{(r)}|\boldsymbol{h}_1)$ represents the likelihood of the sample $x_1$ containing a regional feature $h_1^{(r')} \in \boldsymbol{h}_1$, that is similar to the regional feature $h_2^{(r)}$ in sample $x_2$. Thus, we compute $Q_\Lambda(h_2^{(r)}|\boldsymbol{h}_1)$ as follows.

$$Q_\Lambda(h_2^{(r)}|\boldsymbol{h}_1) = Q_\Lambda(h_2^{(r)}|h_1^{(r')}) = p_{\text{vMF}}\left(h_2^{(r)}\Big|\mu = h_1^{(r')}, \kappa(\|h_2^{(r)}\|)\right), \text{ s.t. } r' = \arg\max_{r'} Q_\Lambda(h_2^{(r)}|h_1^{(r')}) \quad (9)$$

Here, we assume $h_2^{(r)}$ follows a revised vMF distribution in Eq. (3) with mean direction $h_1^{(r')}$, where the $r'$-th region in $x_1$ is selected as the most similar region to the $r$-th region in $x_2$.

As is shown above, the loss $\mathcal{L}_{\text{similarity}}$ enables $h^{(r)}$ to mimic feature representation of $f^{(r)}$ in terms of encoding the sample-wise similarity. Furthermore, we also expect $h^{(r)}$ to reflect the discrimination power of each regional feature. Therefore, we align the regional features $\boldsymbol{h}$ to the coordinate system of $g$ representing the sample-wise discrimination power, in order to represent the regional discrimination power. To this end, we maximize the mutual information between the regional features and the sample features, as the second loss. In other words, this loss enables the discrimination power of regional feature to infer that of the corresponding sample feature.

$$\mathcal{L}_{\text{align}} = -MI(\boldsymbol{h}(X); g(X)) \Rightarrow \frac{\partial \mathcal{L}_{\text{align}}}{\partial \Lambda} = -\mathbb{E}_{Q_\Lambda(\boldsymbol{h}, g)}\left[\frac{\partial \log Q_\Lambda(\boldsymbol{h}|g)}{\partial \Lambda}\right] \quad (10)$$

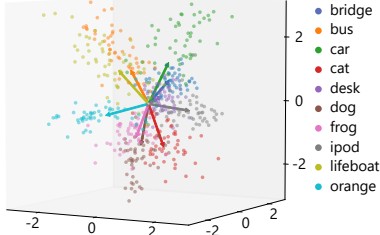

Figure 2: Visualization of sample features learned by VGG-16 in the coordinate system of the Tiny ImageNet categories[1].

| dataset | Tiny ImageNet | | | COCO 2014 | CUB-200-2011 |
|---|---|---|---|---|---|
| DNN | VGG-16 | ResNet-34 | MobileNet-V2 | ResNet-50 | ResNet-34 |
| PCA | -0.65 | -0.78 | -0.81 | -0.56 | -0.78 |
| t-SNE | -0.50 | -0.36 | -0.66 | -0.67 | -0.50 |
| LLE | -0.38 | -0.51 | -0.27 | -0.09 | -0.58 |
| ISOMAP | -0.66 | -0.83 | -0.77 | -0.65 | -0.75 |
| DRPR | -0.87 | -0.89 | -0.88 | -0.77 | -0.77 |
| ours | **-0.94** | **-0.94** | **-0.95** | **-0.84** | **-0.90** |

Table 1: The negative correlation ($\downarrow$) between the visualized sample features' strength and the samples' classification uncertainty.

The joint probability $Q_\Lambda(\boldsymbol{h}, g) = p(g) \cdot \prod_r Q_\Lambda(h^{(r)}|g)^{w^{(r)}}$ reflects the fitness between the discrimination power of a sample and that of its compositional regions. $Q_\Lambda(h^{(r)}|g)$ reflects the fitness between the sample feature $g$ and each $r$-th regional feature $h^{(r)}$, which is assumed to follow a vMF distribution with mean direction $g$, *i.e.* $Q_\Lambda(h^{(r)}|g) = p_{\text{vMF}}(h^{(r)}|g, \kappa')$. In this way, the second loss can be equivalently written as $\mathcal{L}_{\text{align}} = -\mathbb{E}_x[\sum_r w^{(r)} \cdot \cos(g, h^{(r)})]$, where $\kappa'$ has been eliminated (proof in the supplementary material).

In sum, the loss functions in Eq. (7) and (10) enable $h^{(r)}$ to reflect both the feature representation of $f^{(r)}$ and align $h^{(r)}$ to the coordinate system of $g$'s discrimination power. Thus, we learn $\Lambda$ using both losses $\mathcal{L} = \mathcal{L}_{\text{similarity}} + \alpha \cdot \mathcal{L}_{\text{align}}$ ($\alpha > 0$).

**Estimation of each region's importance** $w^{(r)}$. In Eq. (8), we need to estimate the importance of each $r$-th region as $w^{(r)}$. Just like Eq. (7), the objective of estimating $\boldsymbol{w} = [w^{(1)}, ..., w^{(HW)}]$ in each sample is also formulated as the minimization of the KL divergence between $P(x_2|x_1)$ inferred by the DNN and $Q_{\boldsymbol{w}}(x_2|x_1)$ inferred by $f$.

$$\min_{\boldsymbol{w}} KL(P(X_2|X_1)\|Q_{\boldsymbol{w}}(X_2\|X_1)) \Rightarrow \min_{\boldsymbol{w}} \mathbb{E}_{x_1}\left[\sum_{x_2} P(x_2|x_1) \log \frac{P(x_2|x_1)}{Q_{\boldsymbol{w}}(x_2|x_1)}\right] \quad (11)$$

Unlike Eq. (8), we estimate $w$ by formulating $Q_{\boldsymbol{w}}(x_2|x_1)$ using raw features $f$, instead of the projected features $\boldsymbol{h}$, for more accurate estimation. We assume each regional feature $f_2^{(r)}$ contributes independently to $Q_{\boldsymbol{w}}(x_2|x_1)$, *i.e.* $Q_{\boldsymbol{w}}(x_2|x_1) \propto \prod_r Q_{\boldsymbol{w}}(f_2^{(r)}|f_1)^{w_2^{(r)}}$. Then, just like Eq. (9), $Q_{\boldsymbol{w}}(f_2^{(r)}|f_1) = \max_{r'} Q_{\boldsymbol{w}}(f_2^{(r)}|f_1^{(r')})$. In the quantification of $Q_{\boldsymbol{w}}(f_2^{(r)}|f_1^{(r')})$, we further consider the different importance of each channel in $f_2^{(r)}$. To this end, we further estimate the importance of each channel of $f_2$ as $\boldsymbol{v}_2 = [v_2^{(1)}, ..., v_2^{(K)}] \in \mathbb{R}^K$, where $v_2^{(k)} \in \mathbb{R}$ denotes the importance of the $k$-th channel. In this way, we quantify $Q_{\boldsymbol{w}}(f_2^{(r)}|f_1^{(r')})$ as follows.

$$Q_{\boldsymbol{w}}(f_2^{(r)}|f_1^{(r')}) \propto \exp\left[\kappa' \cdot \sum_k \left(v_2^{(k)} \cdot \frac{f_{2,k}^{(r)}}{\|f_2^{(r)}\|_2} \cdot \frac{f_{1,k}^{(r)}}{\|f_1^{(r)}\|_2}\right)\right], \quad (12)$$

where $f_{2,k}^{(r)}$ and $f_{1,k}^{(r)}$ are neural activations of the $k$-th channel in the $r$-th region in $f_2$ and $f_1$, respectively. In our experiments, $\boldsymbol{w}_2$ and $\boldsymbol{v}_2$ were jointly optimized via Eq. (11). For fair comparison between different samples, we force each element in $\boldsymbol{w}_2$ and $\boldsymbol{v}_2$ to be non-negative, and force their $L_1$-norm to be 1. *I.e.* $\boldsymbol{w}_2 \succeq 0$, $\boldsymbol{v}_2 \succeq 0$, $\|\boldsymbol{w}_2\|_1 = 1$, and $\|\boldsymbol{v}_2\|_1 = 1$. This ensures that magnitudes of region's/channel's importance in different samples are similar. Besides, this constraint can also stabilize the optimization process of $\boldsymbol{w}_2$ and $\boldsymbol{v}_2$. The optimization process of $\boldsymbol{w}_2$ and $\boldsymbol{v}_2$ alternates between the following two steps. (i) We first update $\boldsymbol{w}_2$ and $\boldsymbol{v}_2$ via Eq. (11) using the gradient descent method. (ii) We force each element in $\boldsymbol{w}_2$ and $\boldsymbol{v}_2$ to be non-negative and force their $L_1$-norm to be 1, *i.e.* the importance of the $r$-th region $w_2^{(r)}$ is normalized to $\frac{|w_2^{(r)}|}{\|\boldsymbol{w}_2\|_1}$ ($r = 1, ..., HW$), and the importance of the $k$-th channel $v_2^{(k)}$ is normalized to $\frac{|v_2^{(k)}|}{\|\boldsymbol{v}_2\|_1}$ ($k = 1, ..., K$).

### 3.4 Quantifying knowledge points and the ratio of reliable knowledge points

Visualizing the discrimination power of regional features provides us a new perspective to analyze the representation capacity of a DNN, *i.e.* counting knowledge points encoded in different layers, and quantifying the ratio of reliable knowledge points. Up to now, Cheng et al. [7] was the first attempt to quantify the knowledge points encoded in an intermediate layer using the information theory, but

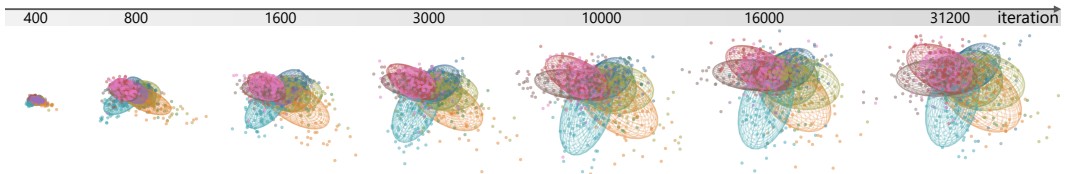

Figure 3: The emergence of regional patterns through the learning process in the coordinate system of visualizing the Tiny ImageNet categories[2].

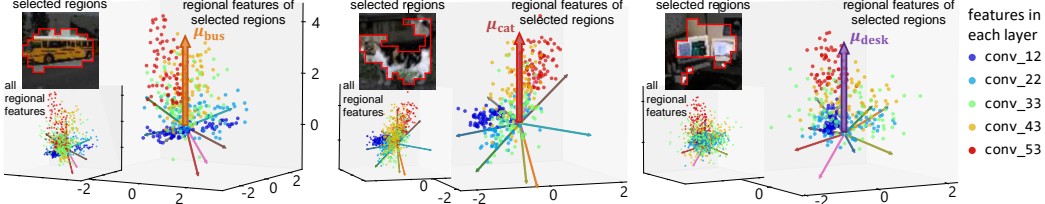

Figure 4: The emergence of regional patterns through the forward propagation in the coordinate system of visualizing the Tiny ImageNet categories[2]. Coordinates along the vertical axis reflect the discrimination power of the target category.

the knowledge points were extracted based on the discard of the pixel-wise information, instead of representing the discrimination power of regional features. In comparison, we quantify the total knowledge points and reliable ones, in terms of their discrimination power. Experiments show that the quantity and quality of knowledge points can well explain knowledge distillation in practice.

Specifically, given a regional feature $h^{(r)}$, if $h^{(r)}$ is discriminative enough for classification of any category, *i.e.* $\max_c p(y = c|h^{(r)}) > \tau$, then we count this regional feature as a *knowledge point*. The classification probability $p(y = c|h^{(r)}) = \frac{\pi_c \cdot \exp[\kappa(\|h^{(r)}\|_2) \cdot \cos(h^{(r)}, \mu_c)]}{\sum_{c'} \pi_{c'} \cdot \exp[\kappa(\|h^{(r)}\|_2) \cdot \cos(h^{(r)}, \mu_{c'})]}$, which is similar to Eq. (6). Furthermore, among all knowledge points, those pushing the classification towards the correct classification, *i.e.* knowledge points satisfying $c^{\text{truth}} = \arg\max_c p(y = c|h^{(r)})$, are taken as *reliable knowledge points*. In this way, *the ratio of reliable knowledge points* is defined as the ratio of reliable knowledge points to total knowledge points, which reflects the quality of visual patterns.

## 4  Experiments

In this section, we used our method to visualize sample features and regional features in VGG-16 [49], ResNet-34/50 [21], MobileNet-V2 [44], which were learned for object classification, based on the Tiny ImageNet dataset [26], the MS COCO 2014 dataset [30], and the CUB-200-2011 dataset [55]. For the MS COCO 2014 dataset and the CUB-200-2011 dataset, we used images cropped by the annotated bounding boxes for both training and testing. Note that the analysis of classification for massive categories requires a large number of category directions in the coordinate system, which will hurt the visualization clarity of the radial distribution. Therefore, to clarify the visualization result, we randomly selected 10 categories from each dataset[2]. Please see the supplementary material for details on the DNNs and datasets.

**Visualization and verification of sample features' discrimination power.** In this experiment, we projected sample features $f$ into a 3-dimensional space (*i.e.* $d'=3$) for visualization. Specifically, we selected the feature before the last fully-connected layer as the sample feature $f$. Figure 2 shows the projected sample features $g$ and each category direction $\mu_c$. The visualization result revealed the semantic similarity between categories. For example, *cat* features were similar to *dog* features, and *bus* features were similar to *lifeboat* features. Besides, the supplementary material shows how the discrimination power of sample features gradually increased through the training process.

Furthermore, in order to examine whether $g$ reflected the discrimination power of sample features, we evaluated the Pearson correlation coefficient between the strength $\|g\|_2$ and the classification uncertainty of each sample $x$. To this end, the classification uncertainty of each sample $x$ was measured as the entropy of its output probability, *i.e.* $H(Y|X = x)$. In Table 1, we compared our method with several visualization methods, such as PCA [38], t-SNE [54], LLE [43], ISOMAP [52], and the

---

[2]For the Tiny ImageNet dataset, we selected *steel arch bridge, school bus, sports car, tabby cat, desk, golden retriever, tailed frog, iPod, lifeboat, and orange*. Please see the supplementary material for other datasets.

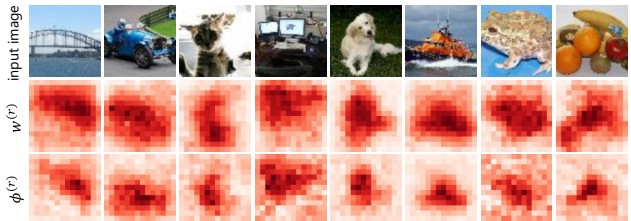

Figure 5: Visualization of the regional importance estimated by our method. Our regional importance is similar to the Shapley value of $f^{(r)}$, which verifies the trustworthiness of our method.

| dataset | DNN | correlation |
|---|---|---|
| Tiny ImageNet | VGG-16 | $0.7707_{\pm 0.16}$ |
| | ResNet-34 | $0.8248_{\pm 0.09}$ |
| | MobileNet-V2 | $0.8169_{\pm 0.13}$ |
| COCO 2014 | ResNet-50 | $0.7572_{\pm 0.18}$ |
| CUB-200-2011 | ResNet-34 | $0.7765_{\pm 0.17}$ |

Table 2: The Pearson correlation coefficient between $\|h^{(r)}\|_2$ and $w^{(r)}$. The feature strength and feature importance were positively related to each other.

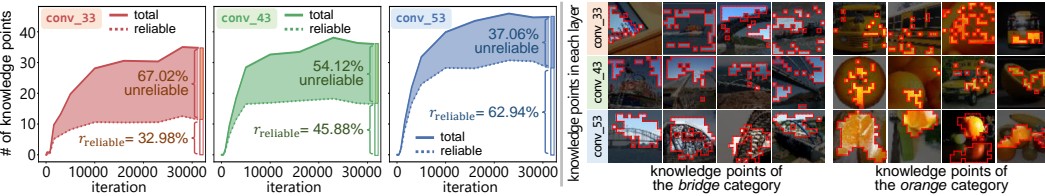

Figure 6: (left) The increase of total knowledge points and reliable knowledge points during training. The ratio of reliable knowledge points, $r_{\text{reliable}} = \#$ of reliable points$/\#$ of all points, increases through the forward propagation. (right) Visualization of image regions of knowledge points towards different categories.

recent DRPR [25]. *Compared with baseline methods, the strength of our projected sample features $g$ was more strongly correlated to the classification uncertainty.* The supplementary material also verified the effectiveness of our method by illustrating contour maps of the classification probability.

**Emergence of discriminative regional features.** Next, we projected regional features $f$ into a 3-dimension space (*i.e.* $d' = 3$) to analyze the importance and reliability of each $f^{(r)}$ towards classification. We set $\alpha = 0.1$. Figure 3 shows the emergence of projected regional features $h^{(r)}$ through the training process, when we selected the output feature of the conv_53 layer of VGG-16 as $f^{(r)}$. The ellipsoid represented the estimated Gaussian distribution of $h^{(r)}$ for image regions cropped from each category. The visualization result showed that the discrimination power and reliability of regional features gradually increased during training. Besides, Figure 4 visualizes regional features extracted from different layers of VGG-16. For clarity, we further selected image regions corresponding to reliable knowledge points in the conv_53 layer. Figure 4 visualizes the selected regions, as well as their regional features. It showed that these regions were not discriminative in low layers, but became discriminative in high layers.

**Visualization and verification of the estimated regional importance.** In this experiment, we estimated regional importance $w^{(r)}$ with $\tilde{\kappa}$ set to 1000. The estimated $w^{(r)}$ was further verified from the following two perspectives. From the first perspective, we compared the estimated regional importance $w^{(r)}$ and the Shapley value [46, 32] $\phi^{(r)}$ of each $r$-th region, when we selected the output feature of the conv_53 layer of VGG-16 as regional features $f^{(r)}$. To this end, the Shapley value $\phi^{(r)}$ was computed as the numerical contribution of $f^{(r)}$ to the DNN output. The Shapley value is the unique unbiased and widely-used [6, 16, 63] metric that fairly allocates the numerical contribution to input features, which satisfies the linearity axiom, the dummy axiom, the symmetry axiom, and the efficiency axiom [2]. Figure 5 shows the high similarity between $w^{(r)}$ and $\phi^{(r)}$ among different regions $r$, which demonstrated the trustworthiness of the estimated regional importance $w^{(r)}$.

Besides, we calculated the Pearson correlation coefficient between the strength of projected features $\|h^{(r)}\|_2$ and their corresponding importance $w^{(r)}$. Table 2 shows the mean value and the standard deviation of the Pearson correlation coefficient through all input samples in each dataset, when we used the output feature of the last convolutional layer as regional features $f$. This proved that feature strength and feature importance were significantly and positively related to each other.

**Quantifying knowledge points and the ratio of reliable knowledge points.** Figure 6(left) shows the increase of knowledge points in different layers through the training of VGG-16. For fair comparison, we normalized the average strength of regional features $\mathbb{E}_{x,r}[\|h^{(r)}\|_2 \text{ given } x]$ in each layer to the average strength of regional features in the conv_53 layer, and therefore we could simply set $\tau = 0.4$. Besides, we also computed the ratio of reliable knowledge points in each layer. Figure 6(left) shows that the ratio of reliable knowledge points in high layers (*e.g.* the conv_53 layer) was higher than that in low layers (*e.g.* the conv_33 layer), which demonstrated the increasing quality of visual

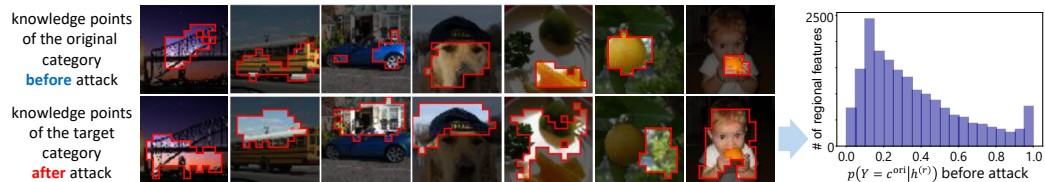

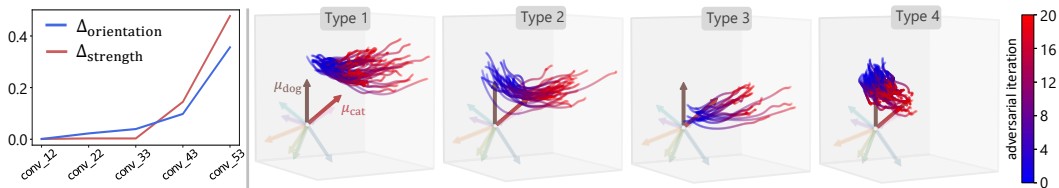

Figure 7: (left) Image regions corresponding to knowledge points in original and adversarial samples. (right) We selected important regions for the target category from adversarial samples, and evaluated the selected regions' utilities/importance of classifying original images to the true category. Most important regions after the attack were not so important before the attack.

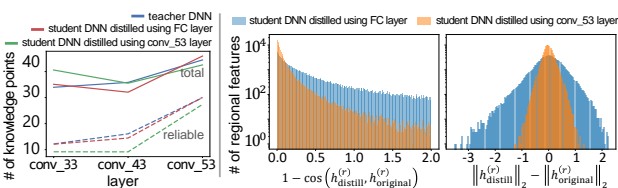

Figure 8: (left) The adversarial attack usually made significant effects on regional features in high layers. (right) Visualization of four types of regional features' trajectories during the attack.

Figure 9: (left) Knowledge distillation caused the DNN to encode less reliable knowledge points. (right) The dissimilarity of features between each student DNN and the teacher DNN, in terms of orientation and strength.

patterns through the forward propagation. Besides, Figure 6(right) highlights the image regions of knowledge points towards different categories. Regional features in high layers were usually more likely to be localized on the foreground than regional features in low layers.

**The adversarial attack mainly affected unreliable regional features in high layers.** We used our method to analyze the change of regional features when we applied the adversarial attack [33] to VGG-16. Given a normal sample $x$, the adversarial sample $x_{\text{adv}} = x + \delta$ was generated via the untargeted PGD attack [33], subject to $\|\delta\|_\infty \leq \frac{1}{255}$. The attack was iterated for $20$ steps with the step size of $\frac{0.1}{255}$. In Figure 7(left), we found that important regions for the classification of the original image (the first row) were usually different from important regions that attacked the classification towards the target category (the second row). More specifically, let $h_{\text{ori}}^{(r)}$ and $h_{\text{adv}}^{(r)}$ denote two corresponding regional features in the same layer before and after the attack. Let us select important regions $\{r\}$ for attacking from adversarial samples, satisfying $p(y = c^{\text{adv}}|h_{\text{adv}}^{(r)}) > 0.4$. Figure 7(right) illustrates the histogram for the selected regions' classification utilities $p(y = c^{\text{ori}}|h_{\text{ori}}^{(r)})$ in the original image. We found that most important regions after the attack were not so important before the attack. Besides, we compared the utility of the attack to regional features in different layers. Let $\Delta_{\text{orientation}} = \mathbb{E}_x[\mathbb{E}_r(\cos(h_{\text{ori}}^{(r)}, h_{\text{adv}}^{(r)}))]$ and $\Delta_{\text{strength}} = \mathbb{E}_x[\mathbb{E}_r(|\|h_{\text{ori}}^{(r)}\|_2 - \|h_{\text{adv}}^{(r)}\|_2|)]$ measure the utility of the attack to regional features' orientation and strength. Figure 8 shows that the adversarial attack mainly affected regional features in high layers, *e.g.* the conv_53 layer in VGG-16. We further categorized all image regions into four types, in terms of their attacking behaviors. To this end, we visualized the trajectories of regional features in the conv_53 during the attack. As Figure 8(right) shows, Type 1 illustrates important image regions for the *dog* category that were directly transferred to the *cat* category without much difficulties. Type 2 illustrates important *dog* regions, in which *dog* features were first damaged and then *cat* features were built up and became important *cat* regions. Type 3 indicates unimportant *dog* regions that were pushed to important *cat* regions. Type 4 indicates important *dog* regions that were damaged by the attack and became unimportant regions.

**The DNN learned via knowledge distillation encoded less reliable visual patterns.** In this experiment, we learned two *student DNN*s (two VGG-16 nets) for knowledge distillation [22]. One *student DNN* was learned by distilling the output feature of the conv_53 layer after the ReLU operation in the *teacher DNN* (a pre-trained VGG-16) to the corresponding layer in the student DNN. The other *student DNN* was learned by distilling the output feature of the penultimate fully-

connected layer after the ReLU operation in the teacher DNN to the corresponding layer in the student DNN. Figure 9(left) compares the number of all knowledge points and reliable knowledge points encoded by the teacher DNN and the two student DNNs, when we quantified knowledge points in `conv_33`/`conv_43`/`conv_53` layers. We found that *student DNNs usually encoded less reliable knowledge points than the teacher DNN.*

• Furthermore, *the student DNN usually learned even less reliable concepts in a layer, if this layer was farther from the target layer used for distillation.* To verify this conclusion, we compared the number of knowledge points between the above two student DNNs. As Figure 9(left) shows, the student DNN distilled using features of the fully-connected layer encoded much less reliable concepts than the student DNN distilled using features of the `conv_53` layer, which verified our conclusion.

• *Although the knowledge distillation could force the student DNN to well mimic features of a specific layer in teacher DNN, there was still a big difference of other layers' regional features between the student DNN and the teacher DNN.* To this end, we evaluated the quality of student DNNs mimicking the teacher DNN. We selected $h_{\text{student}}^{(r)}$ and $h_{\text{teacher}}^{(r)}$ as two corresponding regional features of the student DNN and the teacher DNN in the same layer. Then, we used $1 - \cos(h_{\text{student}}^{(r)}, h_{\text{teacher}}^{(r)})$ and $\|h_{\text{student}}^{(r)}\|_2 - \|h_{\text{teacher}}^{(r)}\|_2$ to measure the difference of orientation and the difference of strength between the two regional features. Figure 9(right) shows the histogram of $1 - \cos(h_{\text{student}}^{(r)}, h_{\text{teacher}}^{(r)})$ and $\|h_{\text{student}}^{(r)}\|_2 - \|h_{\text{teacher}}^{(r)}\|_2$, when we used the `conv_53` layer to evaluate the similarity between the student DNN and the teacher DNN. The similarity between student DNN features and teacher DNN features was lower when the student DNN was distilled based on features in the fully-connected layer (far from the `conv_53` layer), which verified our conclusion.

## 5   Conclusion

In this paper, we propose a method to visualize intermediate visual patterns in a DNN. The visualization illustrates the emergence of intermediate visual patterns in a temporal-spatial manner. The proposed method also enables people to measure the quantity and quality of visual patterns encoded by the DNN, which provides a new perspective to analyze the discrimination power of DNNs. Furthermore, the proposed method provides insightful understanding towards the signal-processing behaviors of existing deep-learning techniques.

## Acknowledgments and Disclosure of Funding

This work is partially supported by the National Nature Science Foundation of China (No. 61906120, U19B2043), Shanghai Natural Science Fundation (21JC1403800,21ZR1434600), Shanghai Municipal Science and Technology Major Project (2021SHZDZX0102).

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
