# Visualizing the Emergence of Intermediate Visual Patterns in DNNs: Supplementary Material

**Mingjie Li**
Shanghai Jiao Tong University
limingjie0608@sjtu.edu.cn

**Shaobo Wang**
Harbin Institute of Technology
181110315@stu.hit.edu.cn

**Quanshi Zhang**[*]
Shanghai Jiao Tong University
zqs1022@sjtu.edu.cn

## A  More visualization of the sample-wise discrimination power

Figure 1 shows the projected sample features $g$ and each category direction $\mu_c$, based on the COCO 2014 dataset and the CUB-200-2011 dataset[1]. The visualization results revealed the semantic similarity between categories. For example, *dining table* features were similar to *pizza* features, and *black footed albatross* features were similar to *laysan albatross* features. Furthermore, Figure 2 shows the projected sample feature $g$ at different iterations of training. This illustrated how the discrimination power of sample features gradually increased through the training process.

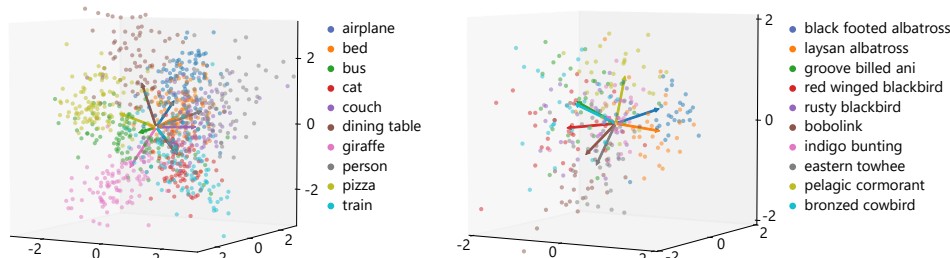

Figure 1: Visualization of sample features learned by (left) ResNet-50 in the coordinate system of visualizing the COCO 2014 categories, and by (right) ResNet-34 in the coordinate system of visualizing the CUB-200-2011 categories[1].

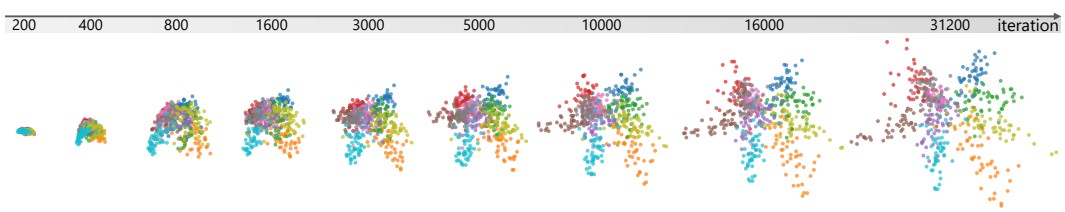

Figure 2: The increasing discrimination power of sample features in VGG-16 during the training process in the coordinate system of visualizing the Tiny ImageNet categories[1].

---

[*]Corresponding author. This work was done under the supervison of Dr. Quanshi Zhang. He is with the John Hopcroft Center and the MoE Key Lab of Artificial Intelligence, AI Institute, at the Shanghai Jiao Tong University, China.

[1]Please see Section G for details of the dataset, and the selection of sample features and regional features.

35th Conference on Neural Information Processing Systems (NeurIPS 2021).

## B  Discussions about the revised vMF distribution

**The vMF distribution** is a kind of distribution for modeling data on a sphere [5, 4, 9]. It is one of the simplest distributions for directional data. Specifically, a vMF distribution on the $(d-1)$-sphere $\mathbb{S}^{d-1}$ in $\mathbb{R}^d$ is parameterized by the mean direction $\mu \in \mathbb{R}^d$, and the concentration parameter $\kappa \geq 0$. Suppose $f \in \mathbb{R}^d$ follows the vMF distribution. Then, the probability density function of $f$ is given by

$$p_{\text{vMF}}(f|\mu,\kappa) = C_d(\kappa) \cdot \exp[\kappa \cdot \cos(\mu, f)], \tag{1}$$

where $C_d(\kappa) = \frac{\kappa^{d/2-1}}{(2\pi)^{d/2} I_{d/2-1}(\kappa)}$ is the normalization constant and $I_{d/2-1}(\cdot)$ denotes the modified Bessel function of the first kind at order $d/2-1$ [1]. Actually, the vMF distribution can be considered as a spherical analogue to the Gaussian distribution on the unit sphere. $\mu$ measures the mean direction. $\kappa$ controls the variance of $f$'s orientation *w.r.t.* the mean direction $\mu$. A large value of $\kappa$ implies a low variance *w.r.t.* $\mu$. In particular, when $\kappa = 0$, the distribution reduces to a uniform distribution on $\mathbb{S}^{d-1}$; when $\kappa \to \infty$, the distribution reduces to a point density. In this way, the probability density of $f$ only depends on its orientation.

**The revised vMF distribution** mentioned in Section 3.1 of the paper takes into account the noise in $f$, *i.e.* $f = f^\star + \epsilon, \epsilon \sim \mathcal{N}(\mathbf{0}, \sigma^2 I_d)$. In this case, all features $f$ of a specific strength $l = \|f\|_2$ have similar vulnerabilities to noises. Features of different strengths have different vulnerabilities to noises. Therefore, the probability density of $f$ not only depends on its orientation but also its strength. In Eq. (3) of the paper, we assume that all features $f$ of a specific strength $l$ follow a vMF distribution with a specific $\kappa(l)$. The concentration parameter $\kappa(l)$ is determined based on statistics of all features of the same strength $l$. Specifically, to quantify $\kappa(l)$, we first sample $\{f_i^\star\}_{i=1}^N$ from $p_{\text{vMF}}(\mu, \kappa)$. Then, the noise $\epsilon \sim \mathcal{N}(\mathbf{0}, \sigma^2 I_d)$ is added to each sample $f_i^\star$, *i.e.* $f_i = f_i^\star + \epsilon$. Since we assume that $f_i$ also follows a vMF distribution, we can estimate $\kappa(l)$ via maximum likelihood estimation (MLE) [13], as follows.

$$\kappa(l) = \arg\max_{\hat{\kappa}} \prod_{i=1}^N p_{\text{vMF}}(f_i|\mu, \hat{\kappa}) \;\Rightarrow\; \kappa(l) = \frac{\|\bar{f}\|_2(d - \|\bar{f}\|_2^2)}{1 - \|\bar{f}\|_2^2}, \tag{2}$$

where $\bar{f} = \frac{1}{N}\sum_i f_i/\|f_i\|_2$. In the calculation of $\kappa(l)$, the sample number $N$ was set to 10000, and $\sigma$ was set to 1.

## C  Derivations on the learning of the mixture model in sample feature visualization via the EM algorithm

This section provides detailed derivations on the learning of the mixture model in Section 3.2 of the paper. In the learning of mixture-model parameters $\{\pi, \mu\} = \{\pi_y, \mu_y\}_{y \in Y}$, we used the EM algorithm to maximize the likelihood $\max_{\{\pi,\mu\}} \prod_g p(g)$. In this way, $\{\pi, \mu\}$ were updated via the following E-step and the M-step.

$$\text{(E-step)} \quad p(y|g) = \frac{\pi_y \cdot \exp\left[\kappa(l_g) \cdot \cos(o_g, \mu_y)\right]}{\sum_{y'} \pi_{y'} \cdot \exp\left[\kappa(l_g) \cdot \cos(o_g, \mu_{y'})\right]}$$

$$\text{(M-step)} \quad \mu_y \propto \mathbb{E}\left[\kappa(l_g) \cdot p(y|g) \cdot o_g\right]_{\text{given } x}, \; \pi_y = \mathbb{E}_x[p(y|g)]_{\text{given } x}, \tag{3}$$

where $l_g = \|g\|_2$ and $o_g = g/l_g$ denote the strength and orientation of $g$, respectively. The derivation is similar to that in [3, 2].

## D  Additional verification of the effectiveness of sample-feature visualization

In this section, we further verify the effectiveness of sample-feature visualization by showing a contour map of the classification probability of the sample feature $g$. In Figure 3, we consider a toy example for the classification of six classes. The red arrow represents the mean direction of the target category, while blue arrows are mean directions of other categories. Figure 3 shows the classification probability towards the target category. We found that sample features $g$ with large strength were more confident towards classification, which further verified the conclusion in Paragraph *visualization and verification of sample features' discrimination power*, Section 4.

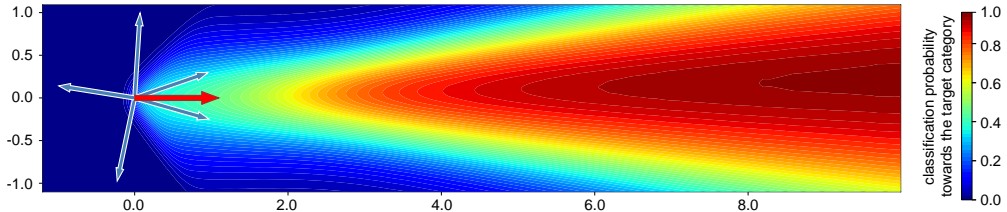

Figure 3: Classification probability towards the target category. The mean direction of the target category is illustrated as the red arrow. We found that sample features $g$ with large strength were usually more confident towards classification.

# E  Derivations of the equivalent form of the loss $\mathcal{L}_{\text{align}}$

This section gives detailed derivations of $\mathcal{L}_{\text{align}} = -\mathbb{E}_x[\sum_r w^{(r)} \cdot \cos(g, h^{(r)})]$ in Section 3.3 of the paper. According to Eq. (10) in the paper, the optimization of $\mathcal{L}_{\text{align}}$ can be written as follows.

$$\frac{\partial \mathcal{L}_{\text{align}}}{\partial \Lambda} = \frac{\partial}{\partial \Lambda} \left[ \mathbb{E}_{Q_\Lambda(\boldsymbol{h})} \left( \log Q_\Lambda(\boldsymbol{h}) \right) - \mathbb{E}_{Q_\Lambda(\boldsymbol{h}, g)} \left( \log Q_\Lambda(\boldsymbol{h}|g) \right) \right], \tag{4}$$

where $Q_\Lambda(\boldsymbol{h})$ is the prior distribution of regional features $\boldsymbol{h}$. For simplicity, we treat $Q_\Lambda(\boldsymbol{h})$ as a constant. Therefore, the optimization can be derived as follows.

$$
\begin{aligned}
\frac{\partial \mathcal{L}_{\text{align}}}{\partial \Lambda} &= -\mathbb{E}_{Q_\Lambda(\boldsymbol{h}, g)} \left[ \frac{\partial \log Q_\Lambda(\boldsymbol{h}|g)}{\partial \Lambda} \right] \\
&= -\mathbb{E}_x \left[ \sum_r w^{(r)} \cdot \frac{\partial \log Q_\Lambda(h^{(r)}|g)}{\partial \Lambda} \right]_{\text{given } x} \\
&= -\mathbb{E}_x \left[ \sum_r w^{(r)} \cdot \frac{\partial \log p_{\text{vMF}}(h^{(r)}|\mu = g, \kappa')}{\partial \Lambda} \right]_{\text{given } x} \\
&= -\mathbb{E}_x \left[ \sum_r w^{(r)} \cdot \frac{\partial \log \frac{1}{C_{d'}(\kappa')} \exp[\kappa' \cdot \cos(g, h^{(r)})]}{\partial \Lambda} \right]_{\text{given } x} \\
&= -\mathbb{E}_x \left[ \sum_r w^{(r)} \cdot \frac{\partial \kappa' \cdot \cos(g, h^{(r)})}{\partial \Lambda} \right]_{\text{given } x} \\
&= -\kappa' \cdot \frac{\partial}{\partial \Lambda} \mathbb{E}_x \left[ \sum_r w^{(r)} \cdot \cos(g, h^{(r)}) \right]_{\text{given } x}
\end{aligned} \tag{5}
$$

$\kappa'$ is a positive constant, which does not essentially affect the convergence of $\Lambda$. Therefore, the loss $\mathcal{L}_{\text{align}}$ can be equivalently written as $\mathcal{L}_{\text{align}} = -\mathbb{E}_x[\sum_r w^{(r)} \cdot \cos(g, h^{(r)})]$.

# F  Discussions about the quantification of knowledge points

This section provides more discussions on the quantification of knowledge points. According to Section 3.4 of the paper, a regional feature is a knowledge point if it is discriminative enough for classification, *i.e.* $\max_c p(y = c|h^{(r)}) > \tau$. Actually, there is a trade off between the value of $\tau$ and the number of knowledge points. If the value of $\tau$ is large, then only a few regional features that are discriminative enough will be quantified as knowledge points. On the other hand, if the value of $\tau$ is small, then a large number of regional features will be quantified as knowledge points. Some of them are not so discriminative. Therefore, we chose $\tau = 0.4$ to balance the trade-off between the discrimination power and the number of knowledge points.

Besides, setting the same value of $\tau$ enables fair comparisons of the discrimination power between features in different layers. First, for each layer, all the $HW$ regional features in $\boldsymbol{h}$ were learned to mimic the sample-wise distribution $P(x_2|x_1)$ inferred by the DNN. Second, for each layer, we uniformly sampled and analyzed the same number of regions. In this way, for each layer, our method used the same number of regional features to mimic the same sample-wise distribution $P(x_2|x_1)$, making the learned regional feature $h^{(r)}$ fairly represent the relative discrimination power of each region, and enabling fair comparisons between regional features through different layers. Furthermore, when quantifying knowledge points in different layers of the DNN, we also normalized the average

strength of regional features $\mathbb{E}_{x,r}[\|h^{(r)}\|_{2 \text{ given } x}]$ in each layer to the same value. This also ensures the fair comparison between regional features in each layer.

## G   Settings of additional experiments in the supplementary material

**Datasets.** We conducted experiments on the task of object classification using the Tiny ImageNet dataset [7], the MS COCO 2014 dataset [8], and the CUB-200-2011 dataset [14]. To clarify the visualization result, we randomly selected ten categories from each dataset. For the Tiny ImageNet dataset, we selected *steel arch bridge (bridge), school bus (bus), sports car (car), tabby cat (cat), desk, golden retriever (dog), tailed frog (frog), iPod, lifeboat*, and *orange* for classification. For the MS COCO 2014 dataset, we selected *airplane, bed, bus, cat, couch, dining table, giraffe, person, pizza*, and *train* for classification. For the CUB-200-2011 dataset, we selected *black footed albatross, laysan albatross, groove billed ani, red winged blackbird, rusty blackbird, bobolink, indigo bunting, eastern towhee, pelagic cormorant*, and *bronzed cowbird* for classification. We used images cropped by the annotated bounding boxes in the MS COCO 2014 dataset and the CUB-200-2011 dataset.

**DNNs, and the selection of sample features and regional features.** We analyzed intermediate-layer features in VGG-16 [12], ResNet-34/50 [6], MobileNet-V2 [11]. We slightly modified the ResNets by changing the stride in `conv_5x` layers to 1. For each of the DNNs, we used the feature before the last fully-connected layer as the raw sample feature. We analyzed regional features in different layers for each DNN. For the VGG-16, we selected the output feature of the `conv_12`, `conv_22`, `conv_33`, `conv_43`, and `conv_53` layers as the raw regional feature. For ResNets, we selected the output feature of the `conv_1`, `conv_2x`, `conv_3x`, `conv_4x`, and `conv_5x` layers as the raw regional feature (denoted as `conv_1`, `conv_2`, `conv_3`, `conv_4`, and `conv_5`). For the MobileNet-V2, we selected the output feature of the 4, 7, 11, 14, 18 layers as the raw regional feature (denoted as `layer_4`, `layer_7`, `layer_11`, `layer_14`, and `layer_18`). For fair comparisons, we downsampled feature maps in different layers of each DNN to the height and width of the output feature at the last convolutional layer. *E.g.*, for intermediate features in VGG-16, feature maps at different layers were downsampled to the size of $14 \times 14$. This makes the learned regional feature $h^{(r)}$ fairly represent the relative discrimination power of each region, thus enabling fair comparisons between regional features through different layers. All our experiments were run using PyTorch 1.7.1 [10] on Ubuntu 18.04, with the Intel(R) Xeon(R) CPU E5-2637 v4 @ 3.50GHz and one NVIDIA(R) GeForce(R) RTX 2080 Ti GPU.