# OpenReview forum: "Visualizing the Emergence of Intermediate Visual Patterns in DNNs"
_NeurIPS.cc/2021/Conference — NeurIPS 2021 Poster_

### Official Review · Reviewer_uv31 · 2021-07-14

**Rating:** 6
**Confidence:** 4

**Summary:**

This paper quantifies the number of discriminative patterns in DNN to evaluate the network capacity through visualization method, and also provides some insights for adversarial attack and knowledge distillation.

**Ethical Concerns:**




**Limitations And Societal Impact:**

In line 295, the paper claims that adv attack mainly affect unreliable regional features in high layers, but lack of full analysis. No experiment shows that the reliable regional features are more robust to attack (or does it mean the reliable region is hard to attack?). Besides, VGG is easy to attack, how is the results for different network (maybe adversarially trained, more robust)? Why these models are hard to attack, do they have less unreliable regional features? I'm also concern about how to apply this method into adversarial attack (sparsely attack the unreliable region) and defense (how to increase the number of reliable regional features).



**Main Review:**

#Originality: The method connects the visual explanation and DNN's discriminative power, which has not been well studied yet.

#Quality: This paper addresses an important problem (evaluate network's discriminative power via visual concepts), and provides a new perspective to analyze the discriminative powers via measure the quantity and quality of visual concepts learned by DNN.

#Clarity: The paper is well-written and easy to follow.

#Significance: The results show some insights for later work, it will be interesting to further explore.

**Time Spent Reviewing:**

3

---

> ### Author Response · Authors · 2021-08-10
> **Response to Reviewer (uv31)**
>
> Thank you very much for your careful review and constructive comments. We try our best to answer all your concerns.
>
> ---
>
> Q1: "No experiment shows that the reliable regional features are more robust to attack (or does it mean the reliable region is hard to attack?)"
>
> A: A good question, but our conclusion is that "most important regions after the attack were not so important before the attack" (Line 303), rather than "reliable regional features are more robust to attack." Here, the word "reliable" is referred as to the correct influence of a regional feature to the classification result, instead of the adversarial robustness.
>
> Nevertheless, we have conducted **an additional experiment** to compare the robustness between reliable knowledge points and unreliable knowledge points. We used the utility of the attack to a regional features' orientation and strength (defined in Line 304-306), i.e. $\Delta_{\text{orientation}}=E_x[E_r(1-\cos(h_{\text{ori}}^{(r)},h_{\text{adv}}^{(r)}))]$ and $\Delta_{\text{strength}}=E_{x}[E_r(|\Vert h_{\text{ori}}^{(r)}\Vert_2-\Vert h_{\text{adv}}^{(r)}\Vert_2|)]$, to measure the robustness of regional features. Smaller values of $\Delta_{\text{orientation}}$ and $\Delta_{\text{strength}}$ indicate higher robustness. In this experiment, we compared the utility of the PGD attack ($\Delta_{\text{orientation}}$ and $\Delta_{\text{strength}}$) between reliable knowledge points and unreliable knowledge points in the conv_53 layer of VGG16 (Line 235). The following table shows that the value of $\Delta_{\text{orientation}}$ on reliable knowledge points was higher than that on unreliable knowledge points. Besides, the value of $\Delta_{\text{strength}}$ on unreliable knowledge points was higher than that on reliable knowledge points. This indicated that for reliable features, the attack mainly affected their reliability. However, for unreliable regional features, the attack had more influence on their importance. Therefore, we could not simply conclude that "reliable regional features are more robust to attack."
>
> |                             | $\Delta_{\text{orientation}}$ | $\Delta_{\text{strength}}$ |
> | --------------------------- | ----------------------------- | -------------------------- |
> | reliable knowledge points   | **0.3214**                    | 0.5833                     |
> | unreliable knowledge points | 0.1624                        | **0.7562**                 |
>
> ---
>
> Q2: "VGG is easy to attack, how is the results for different network (maybe adversarially trained, more robust)?"
>
> A: This is a good question. We have followed your suggestion to conduct **new experiments** on different networks. We analyzed four DNNs trained on the Tiny ImageNet dataset, including the normally trained VGG-16, the normally trained ResNet-34 (Line 235-236), the adversarially trained VGG-16 based on [cite 1], and the distilled VGG-16 (the student DNN distilled based on the output feature of the conv_53 layer in the normally trained VGG-16, following settings in Line 316-317), to analyze the change of regional features' reliability and importance in adversarial attacks. We used $\Delta_{\text{orientation}}=E_x[E_r(1-\cos(h_{\text{ori}}^{(r)},h_{\text{adv}}^{(r)}))]$ and  $\tilde\Delta_{\text{strength}}=E_x\left[E_r\left(\frac{|\Vert {h_{\text{ o r i}}}^{(r)}\Vert_2-\Vert {h_{\text{ a d v}}}^{(r)}\Vert_2|}{\Vert {h_{\text{ o r i}}}^{(r)}\Vert_2}\right)\right]$(Line 304-306) to measure the change of regional features' reliability and importance in the attack, respectively. Note that for fair comparison between different DNNs, we used the strength of regional features $\Vert h_{\text{ori}}^{(r)}\Vert_2$ for normalization when computing the change in regional features' strength $\tilde\Delta_{\text{strength}}$, instead of $\Delta_{\text{strength}}$ in the answer to Q1. In this experiment, we analyzed regional features in the last convolutional layer in each DNN. We calculated $\Delta_{\text{orientation}}$ and $\tilde\Delta_{\text{strength}}$ of the regional features after the attack. The following table compares the change of regional features among the four DNNs after the attack, in terms of their reliability and importance. We found that (1) Compared with the normally trained VGG-16, the attack had less significant effects on regional features in adversarially trained VGG-16. (2) Compared with the normally trained VGG-16, the attack had less significant effects on regional features in the VGG-16 trained via knowledge distillation.
>
> | DNN                              | $\Delta_{\text{orientation}}$ | $\tilde{\Delta}_{\text{strength}}$ |
> | -------------------------------- | ----------------------------- | ---------------------------------- |
> | the normally trained VGG-16      | 0.3091                        | 0.6030                             |
> | the normally trained ResNet-34   | 0.5321                        | 0.7055                             |
> | the adversarially trained VGG-16 | 0.0111                        | 0.4826                             |
> | the distilled VGG-16             | 0.1510                        | 0.2891                             |
>
> [cite 1] Madry, Aleksander, et al. "Towards deep learning models resistant to adversarial attacks." ICLR 2018.
>
> ---
>
> Q3: Ask about the relationship between the difficulty of the attack and the number of unreliable regional features. "Why these models are hard to attack, do they have less unreliable regional features?"
>
> A: A good question. We have followed your suggestion to conduct **an additional experiment** to analyze the relationship between the number of unreliable regional features and the difficulty of the attack. Just like experiments in answers to Q2, we used $\Delta_{\text{orientation}}$ and $\tilde\Delta_{\text{strength}}$ to quantify the change of regional features' reliability and importance in the attack. A smaller value of the change indicates that it is more difficult to attack. Besides, we measure the number of unreliable knowledge points in regional features (denoted as $n_{\text{unreliable}}$). Then, we measured the Pearson correlation coefficient between $\Delta_{\text{orientation}}$ and $n_{\text{unreliable}}$, and the Pearson correlation coefficient between $\tilde\Delta_{\text{strength}}$ and $n_{\text{unreliable}}$ through all input images. The following table shows the correlation calculated based on different DNNs. We found that there was a positive correlation between $\Delta_{\text{orientation}}$ and $n_{\text{unreliable}}$. There was also a positive correlation between $\tilde\Delta_{\text{strength}}$ and $n_{\text{unreliable}}$. This indicated that the decrease of unreliable regional features boosted the difficulty of the attack.
>
> | DNN                              | The correlation between $\Delta_{\text{orientation}}$ and $n_{\text{unreliable}}$ | The correlation between $\tilde\Delta_{\text{strength}}$ and $n_{\text{unreliable}}$ |
> | -------------------------------- | ------------------------------------------------------------ | ------------------------------------------------------------ |
> | the normally trained VGG-16      | 0.7033                                                       | 0.3696                                                       |
> | the normally trained ResNet-34   | 0.6539                                                       | 0.3725                                                       |
> | the adversarially trained VGG-16 | 0.5897                                                       | 0.4057                                                       |
> | the distilled VGG-16             | 0.6477                                                       | 0.3506                                                       |
>
> ---
>
> Q4: "I'm also concern about how to apply this method into adversarial attack and defense."
>
> A: Based on conclusions obtained in the answer to Q3, we can use our method to localize regional features of unreliable knowledge points and penalize the magnitude of these features to boost the adversarial robustness.
>
> Nevertheless, the main contribution of this paper is not to propose new adversarial attack or defense algorithms. Instead, the distinctive contribution of this study is that this paper is a pioneer to bridge the visualization of regional features and the quantitative analysis of regional features' discrimination power. The proposed method is a generic tool and can provide new insights into existing deep-learning techniques, such as adversarial attack and the knowledge distillation.

---

> > ### Comment · Reviewer_uv31 · 2021-08-20
> > **Reply the Paper1341 Authors**
> >
> > I think the authors have addressed a lot of questions in my review with solid experiments. I prefer to increase my score from 5 to 6.
> >
> > Thanks!

---

> > > ### Author Response · Authors · 2021-08-20
> > > **Response to Reviewer (uv31)**
> > >
> > > Thank you very much.

---

### Official Review · Reviewer_kQ6q · 2021-07-16

**Rating:** 7
**Confidence:** 5

**Summary:**

The authors proposed an approach to visualize the regional visual patterns learned by the DNN models during the training. The method shows that (1) adversarial attacks mainly affect the regional features in depth layers. The authors also categorize the adversarial attack impact in four types; and (2) in the distillation scenario, the student learns to extract less reliable knowledge points than the teacher. It is an interesting approach with promising results, quantitative and analysis.


**Ethical Concerns:**

I see no problem.

**Limitations And Societal Impact:**

The authors made no comments about the Limitations And Societal Impact. The authors could address how your research could affect ML applications as possible uses, range of social impacts, and address the positive effects and reduce the downsides of the current approaches.

**Main Review:**

The method divides intermediate layer features into feature components, each of which represents a specific image region. Then, the technique visualizes these feature components' discrimination power and considers discriminative feature components as knowledge points learned by the DNN. Moreover, the authors show how the discrimination power of each visual pattern increases during the learning process and illustrate the effect of using non discriminative patterns in low layers to gradually construct discriminative patterns in high layers during the forward propagation.

I have the following recommendations and suggestions on how this paper can be improved:

(Line 45) "To this end, we believe that bridging regional patterns and a DNN's discrimination power is a more convincing and more intuitive way to reveal the internal behavior of a DNN than mathematical bounds under certain assumptions." The authors need to justify this claim because the paper also made some assumptions, such as lines 126-127 and 147-148.

(Line 214) Can the authors show any quantitative evaluation of this claim? Since the work uses the projected features in other scenarios, it is essential to know how accurate it is.

(Line 245) The input features of fully connected layers are 1D vectors, so what is the shape of this sample feature? Is it (N, 1, 1)? Are the authors using the features before some average pooling layer?

(Lines 266)  How is the mapping between input image regions and the reliable knowledge points made? This is an important step to improve the paper's reproducibility.

(Lines 275-279) The presented qualitative analysis of few examples is interesting, but why not present the quantitative analysis for all images of the dataset?

The quality of the text regions on the Figures should be improved (e.g., Figure 1).

What is the statistical relevance of the results in Figure 6?

What is the impact of the sample confidence in the ratio of reliable knowledge points (RRKP)? Maybe the samples when the model returns the correct answer with high confidence have a higher RRKP than images with low confidence. It is also valid to consider the distillation scenario because the student model may produce low confidence predictions than the teacher.

What is the method performance on out-of-distribution (OOD) samples?

Since distillation can be used as adversarial defense [1], what is the relation between the results of the student model and the adversarial attacks?

There has been little discussion on the limitations of the proposed approach. What are the possible technical limitations? It's not clear how the method deals with noise regions when formulating and estimating the specific importance of different regions. In other words, how accurate is it to assume that all features of a specific region have similar vulnerabilities to noise?

Revise the paper to eliminate the typos in the text (e.g., section 3.1 discrimination as "discimination")

[1] Papernot, Nicolas, et al. "Distillation as a defense to adversarial perturbations against deep neural networks." 2016 IEEE symposium on security and privacy (SP). IEEE, 2016.


**Time Spent Reviewing:**

10

---

> ### Author Response · Authors · 2021-08-10
> **Response to Reviewer (kQ6q) (Part 1)**
>
> Thank you very much for your careful review and constructive comments. We try our best to answer all your concerns.
>
> ---
>
> Q1: About the **paper writing**. The clarification of the claim "(Line 45-47) To this end, we believe that bridging regional patterns and a DNN's discrimination power is a more convincing and more intuitive way to reveal the internal behavior of a DNN than mathematical bounds under certain assumptions."
>
> A: Thank you very much. We agree with that mathematically, the bounds are derived more rigorously. In comparison, our study provides a new perspective to understand the discrimination power of a DNN with more intuitive and more understandable results, such as the visualization of sample discrimination power (Figure 2), the spatial-temporal emergence of regional discrimination power (Figure 3, 4, 6), etc. From this perspective, our method also provides new insights into the analysis of DNNs. Finally, we will polish the language in our paper.
>
> ---
>
> Q2: The clarification of the claim in Line 214. "(Line 214) Unlike Eq. (8), we estimate $w$ by formulating $Q_{{w}}(x_2|x_1)$ using raw features $f$, instead of the projected features ${h}$, for more accurate estimation." "Can the authors show any quantitative evaluation of this claim?"
>
> A: A good question. We have conducted **a new experiment** to verify this claim. In this experiment, we compared the accuracy of the estimated regional importance obtained by the following three different estimation methods.
>
> (1) The first method is to estimate the regional importance based on the raw features $f$, i.e. we directly use the method in Line 210-217 to estimate $w^{(r)}$ as the regional importance.
>
> (2) The second method is to estimate the regional importance based on the projected features $h$. This estimation is the similar as the above estimation of $w^{(r)}$. The only exception is that we replace the sample-wise similarity $Q_w(x_2|x_1)\propto{\prod}_r Q_w (f_2^{(r)}|f_1)^{w_2^{(r)}}$ in Line 216 with $Q_w (x_2|x_1)\propto{\prod}_r Q_w (h_2^{(r)}|h_1)^{w_2^{(r)}}$. We use $\hat{w}^{(r)}$ to denote the regional importance estimated using this method.
>
> (3) The third method is to directly use the classification confidence of a projected regional feature $h^{(r)}$ to its ground-truth category as its regional importance, i.e. $\tilde{w}^{(r)}=\log\frac{p(y\ =\ c^{\text{t r u t h}}\ |\ h^{(r)})}{1\ -\ p(y\ =\ c^{\text{t r u t h}}\ |\ h^{(r)})}$.
>
> In this way, to verify the claim, we aim to prove that the regional importance estimated by the first method is more accurate than the regional importance estimated by the second or the third method. To this end, the accuracy of each regional importance is measured by the Pearson correlation coefficient between the estimated regional importance and Shapley values $\phi^{(r)}$. A higher value of the correlation indicates higher accuracy of the estimated region importance. Given a certain input image, we measured the Pearson correlation coefficient between $w^{(r)}$ and $\phi^{(r)}$, the Pearson correlation coefficient between $\hat{w}^{(r)}$ and $\phi^{(r)}$, and the Pearson correlation coefficient between $\tilde{w}^{(r)}$ and $\phi^{(r)}$, over different regional features. For each estimation method, we averaged the correlation through all input images. The table below shows that the correlation between $w^{(r)}$ and $\phi^{(r)}$ was higher than the correlation between $\hat{w}^{(r)}$ and $\phi^{(r)}$, and the correlation between  $\tilde{w}^{(r)}$ and $\phi^{(r)}$. This showed the high accuracy of estimating regional importance based on raw features $f$. This verified our claim that estimating regional importance based on raw features $f$ was more accurate than estimating regional importance based on projected features $h$.
>
> | Dataset       | DNN          | The correlation between $w^{(r)}$ and $\phi^{(r)}$ | The correlation between $\hat{w}^{(r)}$ and $\phi^{(r)}$ | The correlation between $\tilde{w}^{(r)}$ and $\phi^{(r)}$ |
> | ------------- | ------------ | -------------------------------------------------- | -------------------------------------------------------- | ---------------------------------------------------------- |
> | Tiny ImageNet | ResNet-34    | **0.8943**                                         | 0.3638                                                   | 0.6538                                                     |
> | Tiny ImageNet | VGG-16       | **0.6307**                                         | 0.2836                                                   | 0.5428                                                     |
> | Tiny ImageNet | MobileNet-V2 | **0.8658**                                         | 0.4361                                                   | 0.7301                                                     |
> | COCO 2014     | ResNet-50    | **0.8814**                                         | 0.3881                                                   | 0.4406                                                     |
> | CUB-200-2011  | ResNet-34    | **0.8561**                                         | 0.3835                                                   | 0.3846                                                     |
>
> ---
>
> Q3: "The input features of fully connected layers are 1D vectors, so what is the shape of this sample feature? Is it (N, 1, 1)? Are the authors using the features before some average pooling layer?"
>
> A: We selected the feature before the last fully-connected layer to be the sample feature, which is after the average pooling layer. This is a 1D vector with N dimensions. Please see Line 246 for details. We will revise our paper to clarify this.
>
> ---
>
> Q4: Ask for the quantitative analysis of the relationship between the estimated regional importance $w^{(r)}$ and the Shapley value $\phi^{(r)}$. "The presented *qualitative* analysis of few examples is interesting, but why not present the *quantitative* analysis for all images of the dataset?"
>
> A: A good suggestion. We have followed your suggestion to design a quantitative metric to analyze the relationship between $w^{(r)}$ and $\phi^{(r)}$. Given a certain input image, we measured the Pearson correlation coefficient between $w^{(r)}$ and $\phi^{(r)}$ over different regional features. We conducted **an additional experiment** to analyze their correlation by evaluating the correlation between $w^{(r)}$ and $\phi^{(r)}$ through all input images for each DNN. The table below shows that there was a positive relationship between $w^{(r)}$ and $\phi^{(r)}$. This demonstrated that the estimated importance $w^{(r)}$ could objectively reflect the importance of each region.
>
> | Dataset                                                      | Tiny ImageNet                  | Tiny ImageNet                  | Tiny ImageNet                  | COCO 2014                      | CUB-200-2011                   |
> | ------------------------------------------------------------ | ------------------------------ | ------------------------------ | ------------------------------ | ------------------------------ | ------------------------------ |
> | DNN                                                          | ResNet-34                      | VGG-16                         | MobileNet-V2                   | ResNet-50                      | ResNet-34                      |
> | The Pearson correlation coefficient between $w^{(r)}$ and $\phi^{(r)}$ through all images | $0.8943{\scriptsize\pm0.0994}$ | $0.6307{\scriptsize\pm0.1831}$ | $0.8658{\scriptsize\pm0.1432}$ | $0.8814{\scriptsize\pm0.1623}$ | $0.8561{\scriptsize\pm0.1680}$ |
>
> ---
>
> Q5: About the **image quality**. "The quality of the text regions on the Figures should be improved (e.g., Figure 1)."
>
> A: Thanks for pointing this out. We will follow your suggestion to improve the quality of figures (e.g. enlarge text sizes and make them clearer).
>
> ---
>
> Q6: "How is the mapping between input image regions and the reliable knowledge points made? This is an important step to improve the paper's reproducibility."
>
> A: A good question. We will revise our paper to further clarify the mapping between image regions and the reliable knowledge points. First of all, as is discussed in [64], the actual receptive fields of intermediate-layer features in a DNN are usually complex. However, based on experience of massive experiments, we can simplify the computation of the receptive fields as follows. Let us consider an intermediate-layer feature $f\in\mathbb{R}^{K\times H\times W}$, which is composed of $HW$ regional features $f^{(r)}\in\mathbb{R}^K,(r=1,2,...,HW)$. Then, we can simply divide the input image into $H\times W$ grids, and the $r$-th region in the input can be roughly considered as the receptive field of the $r$-th regional feature $f^{(r)}$. Therefore, if a regional feature $f^{(r)}$ is quantified as a reliable knowledge point, then we can roughly consider the DNN encodes reliable knowledge in the $r$-th image region. The visualization in this paper only points out the position of reliable knowledge points, but it does not represent the exact receptive fields of these knowledge points.

---

> ### Author Response · Authors · 2021-08-10
> **Response to Reviewer (kQ6q) (Part 2)**
>
> Q7: "What is the statistical relevance of the results in Figure 6?"
>
> A: We would like to clarify the results in Figure 6, though we are very sorry that we cannot fully understand this question. The statistics in Figure 6(left) shows the ratio of reliable knowledge points $r_{\text{reliable}}=\frac{\verb|#|\text{ of reliable knowledge points}}{\verb|#|\text{ of all knowledge points}}$ and the ratio of unreliable knowledge points $\frac{\verb|#|\text{ of unreliable knowledge points}}{\verb|#|\text{ of all knowledge points}}$ in each layer of the trained DNN. We found that the ratio of reliable knowledge points increased through the forward propagation. This demonstrated the increasing quality of visual patterns through the forward propagation.
>
> ---
>
> Q8: "What is the impact of the sample confidence in the ratio of reliable knowledge points (RRKP)?"
>
> A: This is a good question. We have conducted **a new experiment** to show the positive relationship between sample classification confidence and the ratio of reliable knowledge points (RRKP). The sample classification confidence is quantified as the $\log\frac{p(y\ =\ c^{\text{t r u t h}}\ |\ x)}{1\ -\ p(y\ =\ c^{\text{t r u t h}}\ |\ x)}$. We measured the Pearson correlation coefficient of sample classification confidence and RRKP using the output feature of the last convolutional layer for each DNN. The following shows a positive correlation between sample classification confidence and RRKP, which indicates a positive relationship between the sample classification confidence and RRKP.
>
> | Dataset                                                      | Tiny ImageNet | Tiny ImageNet | Tiny ImageNet | COCO 2014 | CUB-200-2011 |
> | ------------------------------------------------------------ | ------------- | ------------- | ------------- | --------- | ------------ |
> | DNN                                                          | ResNet-34     | VGG-16        | MobileNet-V2  | ResNet-50 | ResNet-34    |
> | The Pearson correlation coefficient between sample classification confidence and RRKP | 0.4114        | 0.4828        | 0.4967        | 0.4039    | 0.6523       |
>
> ---
>
> Q9: "What is the method performance on out-of-distribution (OOD) samples?"
>
> A: A good question. We have followed your suggestion to conduct **a new experiment** to evaluate the method performance on OOD samples. Specifically, in the current problem setting, OOD samples are considered as adversarial samples obtained via the PGD attack (Line 296-297). The method performance is quantified as the value of $KL[P(X_2|X_1)\Vert Q_{\Lambda}(X_2|X_1)]$ (in Eq. (7)), which measures how well the projected regional features $h^{(r)}$ reflects sample-wise similarities. A smaller KL divergence indicates better performance. Thus, if the value of $KL[P(X_2|X_1)\Vert Q_{\Lambda}(X_2|X_1)]$ on normal samples is similar to that on OOD samples, we can consider the method performance is good on OOD samples.
>
> In this way, let us revisit the experiment. We compared the method performance between normal samples and OOD samples. OOD samples were used in the learning of the projection matrix $\Lambda$, and we calculated the value of $KL[P(X_2|X_1)\Vert Q_{\Lambda}(X_2|X_1)]$ based on the conv_53 layer feature in VGG-16 for normal samples and OOD samples. The table below shows that the value of $KL[P(X_2|X_1)\Vert Q_{\Lambda}(X_2|X_1)]$ on normal samples was similar to that on OOD samples. This indicated that the method performance on OOD samples was good.
>
> |                                                      | on normal samples | on OOD samples |
> | ---------------------------------------------------- | ----------------- | -------------- |
> | $KL[P(X_2\vert X_1)\Vert Q_{\Lambda}(X_2\vert X_1)]$ | 0.8491            | 0.8619         |
>
> ---
>
> Q10: About the relationship between knowledge distillation and adversarial attack. "Since distillation can be used as adversarial defense [cite 1], what is the relation between the results of the student model and the adversarial attacks?"
>
> A: A good question. We have followed your suggestion to conduct **a new experiment** to compare the distilled student model with the normal model, in terms of the change of regional features in their reliability and importance after the attack (Line 304-306). The student model was a VGG-16 net (termed the distilled VGG-16), which was learned by pushing the output feature of its conv_53 layer towards the corresponding feature in a normally trained VGG-16 (termed the original VGG-16, Line 235) for distillation. We used $\Delta_{\text{orientation}}=E_x[E_r(1-\cos(h_{\text{ori}}^{(r)},h_{\text{adv}}^{(r)}))]$ to measure the utility of the attack to regional features' orientation, which reflects the change of regional features' reliability. Besides, we used  $\tilde\Delta_{\text{strength}}=E_x\left[E_r\left(\frac{|\Vert {h_{\text{ o r i}}}^{(r)}\Vert_2-\Vert {h_{\text{ a d v}}}^{(r)}\Vert_2|}{\Vert {h_{\text{ o r i}}}^{(r)}\Vert_2}\right)\right]$to measure the utility of the attack to regional features' strength, which reflects the change of regional features' importance. Note that for fair comparison between different DNNs, we used the strength of regional features $\Vert h_{\text{ori}}^{(r)}\Vert_2$ for normalization when computing the change in regional features' strength $\tilde\Delta_{\text{strength}}$, instead of $\Delta_{\text{strength}}$ in Line 305. Larger values of $\Delta_{\text{orientation}}$ and $\tilde\Delta_{\text{strength}}$ indicates more significant changes in reliability and importance. The following table shows $\Delta_{\text{orientation}}$ and $\tilde\Delta_{\text{strength}}$ of regional features in the original VGG-16 and the distilled VGG-16. We found that the utilities of the attack on the original VGG-16 was higher than those on the distilled VGG-16. This indicated that the regional features were more robust in the distilled DNN than those in the normally trained DNN.
>
> |                      | $\Delta_{\text{orientation}}$ | $\tilde\Delta_{\text{strength}}$ |
> | -------------------- | ----------------------------- | -------------------------------- |
> | the original VGG-16  | 0.3091                        | 0.6030                           |
> | the distilled VGG-16 | 0.1510                        | 0.2891                           |
>
> [cite 1] Papernot, Nicolas, et al. "Distillation as a defense to adversarial perturbations against deep neural networks." 2016 IEEE symposium on security and privacy (SP). IEEE, 2016.
>
> ---
>
> Q11: The lack of discussions on the limitation. "There has been little discussion on the limitations of the proposed approach".
>
> A: Thanks. We would like to discuss the limitations here, and we will also add them to the paper. Our research only focuses on  image classification tasks based on convolutional neural networks (CNNs), and we analyze the discrimination power of regional features in the CNN. We cannot evaluate the word discrimination power in natural language processing (NLP) tasks, since intermediate-layer features usually describe the entire input sentence, instead of a specific word.
>
> ---
>
> Q12: Concerns about the accuracy of the assumption that "all features $f$ have similar vulnerabilities to noises".
>
> A: A good question. We have conducted **a new experiment** to empirically show that features corresponding to each region have similar vulnerabilities to noise. In this paper, the vulnerability of a regional feature to noise is quantified as the feature's sensitivity to noise. Specifically, given a trained DNN, let $f\in\mathbb{R}^{K\times W\times H}$ be the intermediate-layer feature tensor of an input image $x$. When we add a Gaussian noise to $x$, i.e. $x'=x+\epsilon,\ \epsilon\sim\mathcal{N}(0,\sigma^2 I)$, the feature of $x'$ becomes $f'$. Then, the sensitivity of the $r$-th regional feature vector $f\in\mathbb{R}^K$ to noise is quantified as $\Vert f^{(r)}-f'^{(r)}\Vert_2$.
>
> To verify the assumption, we evaluated whether the sensitivities of different regional features were similar. To this end, we averaged the above sensitivity over different inputs and noises, i.e. $\Delta^{(r)}=\mathbb{E}_x\mathbb{E}_\epsilon[\Vert f^{(r)}-f'^{(r)}\Vert_2]$, to measured the vulnerability of the $r$-th regional feature to noise. We conducted this experiment based on five DNNs below. The table below shows the mean value and standard deviation of $\Delta^{(r)}$ over different regions, i.e. $mean=\mathbb{E}_r[\Delta^{(r)}]$ and $std=\sqrt{\mathbb{E}_r[(\Delta^{(r)}-mean)^2]}$. We found that different regional features had similar sensitivities to noise, so that the sensitivities of different regional features are all close to the mean sensitivity. This was verified by the phenomenon that $std\ll mean$. Thus, we verified the assumption that "all features $f$ have similar vulnerabilities to noises."
>
> | dataset                                                      | Tiny ImageNet                  | Tiny ImageNet                  | Tiny ImageNet                  | COCO 2014                      | CUB-200-2011                    |
> | ------------------------------------------------------------ | ------------------------------ | ------------------------------ | ------------------------------ | ------------------------------ | ------------------------------- |
> | DNN                                                          | ResNet-34                      | VGG-16                         | MobileNet-V2                   | ResNet-50                      | ResNet-34                       |
> | the mean value and standard deviation of $\Delta^{(r)}$ over different regions $r$ | $3.5944{\scriptsize\pm0.3866}$ | $2.9176{\scriptsize\pm0.2077}$ | $4.7463{\scriptsize\pm0.2454}$ | $3.4236{\scriptsize\pm0.4502}$ | $10.5706{\scriptsize\pm0.8646}$ |
>
> ---
>
> Q13: Typos in this paper, "e.g., section 3.1 discrimination as 'discimination' "
>
> A: Thank you. We will fix this typo.

---

> > ### Comment · Reviewer_kQ6q · 2021-08-22
> > **Response to authors rebuttal**
> >
> > Thank you for the comments and the new experiments. Congratulations on the paper.

---

> > > ### Author Response · Authors · 2021-08-22
> > > **Response to Reviewer (kQ6q)**
> > >
> > > Thank you very much.

---

### Official Review · Reviewer_naDB · 2021-07-17

**Rating:** 7
**Confidence:** 4

**Summary:**

The authors present an interpretability method that shows the regional visual patterns (ie, knowledge points) that are important for the discriminative power of a network on a given image. They also explore this in the context of model distillation and adversarial attacks


**Main Review:**

**Originality**
This paper definitely seems original to me. It is an interesting method that goes beyond just looking at image pieces that are relevant for a given prediction, to try and match this with the discriminative power of the network.

**Quality**
I believe this is a high quality submission. The methods are sound (if a bit convoluted-- see clarity section below), and the analysis of model distillation and adversarial attacks is really compelling


**Clarity**
Most of the paper is clear, but I found section 3 a bit dense and difficult to understand. I know part of this is inherent since this section is math-heavy, but it could also really benefit from another diagram similar to figure 1 or figure 3 to just explain at a high level what the intuition is for the algorithm, before deeply diving into the derivations.

Also, I'm still a bit confused about algorithm and want to make sure I'm understanding it correctly. Is this correct?
- Chunk the image into pieces (ie, geographic regions-- might be a cat ear, section of the wall, etc)
- Run each of these image pieces through the CNN to get its embedding at each layer
- Get the dot product of the image piece embeddings with feature space directions for each class
- For these dot products, high value = high discriminative power = knowledge point
- To make visualization, show the regions that had high dot products

If this is correct, section 3 probably doesn't need to be as long (parts could be moved to the appendix). That being said, I also definitely might be missing something more complicated about the method!

Also, one typo:
>while others aimed to generate an input image that cause high
(cause => causes)

**Significance**
This is an interesting and novel take on visualizing image regions that are relevant to DNN predictions.


**Time Spent Reviewing:**

45 min

---

> ### Author Response · Authors · 2021-08-10
> **Response to Reviewer (naDB)**
>
> Thank you very much for your careful review and constructive comments. We try our best to answer all your concerns.
>
> ---
>
> Q1: A suggestion for **paper writing**. "it could also really benefit from another diagram similar to figure 1 or figure 3 to just explain at a high level what the intuition is for the algorithm, before deeply diving into the derivations."
>
> A: Thank you. This is a good suggestion. We will follow your suggestion and add a diagram to improve the readability. In this diagram, we will illustrate the radial distribution in Eq. (3), the mixture model in Eq. (5), how each regional feature contributes to sample-wise similarity in Eq. (8), etc.
>
> ---
>
> Q2: About the paper writing. "I'm still a bit confused about algorithm and want to make sure I'm understanding it correctly. ... "
>
> A: Basically, your understanding is correct.
>
> - "Chunk the image into pieces (i.e., geographic regions -- might be a cat ear, section of the wall, etc)." It is partially correct. We would like to revise the paper to further clarify that we divide the intermediate-layer features into different regional features, instead of directly dividing the image into geographic regions. Please see Line 156-161 for details.
> - "Run each of these image pieces through the CNN to get its embedding at each layer." We do not feed the image pieces one-by-one into the CNN. Instead, we directly compute the feature tensor of a convolutional layer given the entire image. Then, we divide the feature tensor into different regional feature vectors, and project these regional features into a low dimensional space for visualization.
> - "Get the dot product of the image piece embeddings with feature space directions for each class." It is correct.
> - "For these dot products, high value = high discriminative power = knowledge point." It is correct. Please see Line 227-232 for details on the quantification of knowledge points.
> - "To make visualization, show the regions that had high dot products." It is correct. Figure 3 visualizes both knowledge points and other regional features. Only knowledge points have "high dot products." Figure 6(right) visualizes knowledge points.
>
> ---
>
> Q3: About the paper writing. "... section 3 probably doesn't need to be as long (parts could be moved to the appendix)."
>
> A: Thank you. We will follow your suggestion to move parts of the mathematical derivation into the appendix.
>
> ---
>
> Q4: A typo: "while others aimed to generate an input image that cause high (cause => causes)"
>
> A: Thank you. We will fix this typo.

---

> > ### Comment · Reviewer_naDB · 2021-08-30
> > **Response to authors**
> >
> > Thanks for the updates and explanation, and nice work on the paper

---

> > > ### Author Response · Authors · 2021-08-31
> > > **Response to Reviewer (naDB)**
> > >
> > > Thank you very much.

---

### Official Review · Reviewer_et3m · 2021-07-20

**Rating:** 7
**Confidence:** 3

**Summary:**

Paper proposes a visualization method for probing into the intermediate visual patterns learned by the DNN layers. The idea is to learn a linear transformation to project the original DNN features into a lower dimensional space for mimicing the classification based on the original features. In other words, the projected low-dim features infer the similarity among samples. Experiments are conducted multiple datasets and DNN architectures.

**Limitations And Societal Impact:**

I do not see potential -ve societal impact of this work.

**Main Review:**

- Experiments reveal that the low dimensional projections preserve semantic similarity. Also, the magnitude of the projections is shown to be correlated (better than baselines and existing works) to the confidence of classification. Similarly, in case of regional features also, they show that both the discriminative ability and reliability improve temporally with training.
- Regional importance estimated by the proposed approach is shown to be aligned with the Shapeley value (contribution of that region to the DNN o/p).
- Paper presents multiple use cases of the proposed visualization along with understanding the DNN learnings and verification of the known notions.
- Proposed approach infers that the adversaries majorly affect the easy to attack (or, unreliable) regional features from the semantic (higher) layers.
- Similarly, interesting findings are inferred by the paper about the process of knowledge distillation (although the faithfulness of the metrics such as #knowledge points that led to these inferences has not been studied rigorously).
- In summary, the paper presents a different (and seemingly complex) method for going into the learning aspects of the DNNs. However, it sort of gives a feeling that multiple inferences are not strong novel findings (yet they may be considered for adding reliability on the proposed method). For instance, evolution of discriminative features with training, increase in the discriminative ability of the features (and number of reliable knowledge points) towards the deeper layers, etc. In this regard, authors' attempt to explain phenomena such as adversarial attacks and knowledge distillation helps.

Minor

- line 164: 'needs two overcome to challenges' -> 'needs to overcome two challenges'

**Time Spent Reviewing:**

~6

---

> ### Author Response · Authors · 2021-08-10
> **Response to Reviewer (et3m)**
>
> Thank you very much for your careful review and constructive comments. We try our best to answer all your concerns.
>
> ---
>
> Q1: "Similarly, interesting findings are inferred by the paper about the process of knowledge distillation (although the faithfulness of the metrics such as #knowledge points that led to these inferences has not been studied rigorously)."
>
> A: Thank you. We have followed your suggestion to discuss the faithfulness of the quantification of knowledge points both theoretically and experimentally.
>
> First, theoretically, according to $Q_\Lambda(x_2|x_1)\propto\prod_r Q_\Lambda(h_2^{(r)}|h_1)^{w_2^{(r)}}$ in Line 186, each projected regional feature $h^{(r)}$ contributes to the sample-wise similarity. According to $Q_\Lambda (h_2^{(r)}|h_1)=Q_\Lambda(h_2^{(r)}|h_1^{(r')})=p_{\text{vMF}}(h_2^{(r)}|\mu=h_1^{(r')},\kappa(\Vert h_2^{(r)}\Vert))$ in Eq. (9), the strength $\Vert h_2^{(r)}\Vert$ controls the $\kappa$ value, thereby reflecting the **impact/importance** of $h_2^{(r)}$ to the sample-wise similarity. Besides, $Q_\Lambda(h,g)=p(g)\cdot \prod_r Q_\Lambda(h^{(r)}|g)^{w^{(r)}}$ in Line 201 shows that each $h^{(r)}$ contributes to the sample discrimination power. According to $Q_\Lambda(h^{(r)}|g)=p_{\text{vMF}}(h^{(r)}|\mu=g,\kappa')$ in Line 204, the orientation of $h^{(r)}$ reflects the **reliability** of representing $g$'s discrimination power. Therefore, in the distribution $p_{\text{vMF}}(h^{(r)}|\mu_c,\kappa(\Vert h^{(r)}\Vert))$, **the strength of the projected regional feature $\Vert h^{(r)}\Vert_2$ reflects the importance of the $r$-th regional feature, and the orientation of $h^{(r)}$ represents its reliability of the classification towards category $c$.** In other words, $h^{(r)}$ can reflect the discrimination power of regional features.
>
> In this way, we can measure the classification probability towards category $c$ as follows.
> $$
> p(y=c|h^{(r)})=\frac{p(y=c)p(h^{(r)}|y=c)}{\sum_{c'}p(y=c')p(h^{(r)}|y=c')}=\frac{p(y=c)p_{\text{vMF}}(h^{(r)}|\mu_c,\kappa(\Vert h^{(r)}\Vert))}{\sum_{c'}p(y=c')p_{\text{vMF}}(h^{(r)}|\mu_{c'},\kappa(\Vert h^{(r)}\Vert))}=\frac{\pi_c\cdot\exp[\kappa(\Vert h^{(r)}\Vert)\cdot\cos(h^{(r)}, \mu_c)]}{\sum_{c'}\pi_{c'}\cdot\exp[\kappa(\Vert h^{(r)}\Vert)\cdot\cos(h^{(r)}, \mu_{c'})]}
> $$
> If $h^{(r)}$ is discriminative enough for classification of any category, i.e. the classification probability of $h^{(r)}$ is larger than a certain threshold, $\max_c p(y=c|h^{(r)})>\tau$, we can consider the $r$-th regional feature as a knowledge point.
>
> Second, we also examine the faithfulness of knowledge points using the following experiments. The basic idea is that if knowledge points have much more influence on the classification than other regional features, we can consider that the quantification of knowledge points is faithful. Therefore, we have conducted **additional experiments** to compare the influence on classification between knowledge points and other regional features.
>
> The influence can be measured in the following two ways. (1) The larger Shapley value $\phi^{(r)}$ (Line 275-278) of a regional feature indicates that it has stronger influence on classification.  If the Shapley values of knowledge points are higher than those of other regional features, then it verifies that the quantification of knowledge points is faithful. (2) The larger decrease of the DNN's classification confidence (when we mask a regional feature) also indicates that it has stronger influence on classification. Here, the classification confidence is quantified as $\log\frac{p(y\ =\ c^{\text{t r u t h}}\ |\ x)}{1\ -\ p(y\ =\ c^{\text{t r u t h}}\ |\ x)}$. If the decrease corresponding to knowledge points is larger than the decrease corresponding other regional features, then it also verifies that the quantification of knowledge points is faithful.
>
> 1. In the first experiment, we compared Shapley values between knowledge points and other regional features. We computed Shapley values of knowledge points and other regional features extracted from the last convolutional layer in VGG-16, ResNet-34/50 and MobileNet-V2 (Line 235-237). We averaged the Shapley values over different input images. The following table shows Shapley values corresponding to knowledge points and other regional features. We found that the Shapley values of knowledge points were higher than those of other regional features. This verified the faithfulness of the quantification of knowledge points.
>
>    | Dataset       | DNN          | The Shapley value of knowledge points | The Shapley value of other regional features |
>    | ------------- | ------------ | ------------------------------------- | -------------------------------------------- |
>    | Tiny ImageNet | ResNet-34    | $\bf{7.3068\times10^{-2}}$            | $2.4459\times10^{-2}$                        |
>    | Tiny ImageNet | VGG-16       | $\bf{6.8963\times10^{-2}}$            | $2.2088\times10^{-2}$                        |
>    | Tiny ImageNet | MobileNet-V2 | $\bf{1.5010\times10^{-1}}$            | $0.5412\times10^{-1}$                        |
>    | COCO 2014     | ResNet-50    | $\bf{1.3845\times10^{-1}}$            | $0.4952\times10^{-1}$                        |
>    | CUB-200-2011  | ResNet-34    | $\bf{1.3723\times10^{-1}}$            | $0.4977\times10^{-1}$                        |
>
> 2. In the second experiment, we compared the decrease of the DNN's classification confidence when we masked a knowledge point or another regional feature. Specifically, we masked regional features by setting them to zero, and used the masked feature for classification.  Just like the first experiment, we used the above five DNNs. The following table shows the change of the classification confidence when we masked a knowledge point or another regional feature. We averaged the value of changes over different input images. We found that the classification confidence decreased more significantly when we masked a knowledge point, compared with the case when we masked another regional feature. This also verified the quantification of knowledge points was faithful.
>
>      | Dataset       | DNN          | The change of classification confidence when masking a knowledge point | The change of classification confidence when masking another regional feature |
>    | ------------- | ------------ | ------------------------------------------------------------ | ------------------------------------------------------------ |
>    | Tiny ImageNet | ResNet-34    | $\bf{-4.2719\times10^{-2}}$                                  | $-1.2971\times10^{-2}$                                       |
>    | Tiny ImageNet | VGG-16       | $\bf{-4.4111\times10^{-2}}$                                  | $-1.0258\times10^{-2}$                                       |
>    | Tiny ImageNet | MobileNet-V2 | $\bf{-2.0407\times10^{-1}}$                                  | $-0.6242\times10^{-1}$                                       |
>    | COCO 2014     | ResNet-50    | $\bf{-9.3916\times10^{-3}}$                                  | $-3.3779\times10^{-3}$                                       |
>    | CUB-200-2011  | ResNet-34    | $\bf{-4.7260\times10^{-2}}$                                  | $-1.4876\times10^{-2}$                                       |
>
> Furthermore, we have also discussed the faithfulness of the quantification of knowledge points in the supplementary material. Please see Section F in the supplementary material for details.
>
> ---
>
> Q2: "It sort of gives a feeling that multiple inferences are not strong novel findings (yet they may be considered for adding reliability on the proposed method)."
>
> A: Some of our experiments just show the emergence of intermediate visual patterns in a spatial-temporal manner, but these experiments are just designed to verify the effectiveness of our visualization method.
>
> Instead of simply visualizing above "not strong novel findings," the distinctive contribution of this study is that this paper is a pioneer to bridge the visualization of regional features and the quantitative analysis of regional features' discrimination power. Specifically, we have made breakthroughs in the following three directions. (1) We propose a generic method to visualize the emergence of intermediate visual patterns in a temporal-spatial manner. (2) We quantify the knowledge points in intermediate-layer features based on the visualization. (3) The proposed method also provides new insights into existing deep-learning techniques, such as adversarial attack and the knowledge distillation. Please see Line 29-38 and Line 48-55 for more details. If this paper can be accepted, we will add more discussion about this in the final version.
>
> ---
>
> Q3: A typo: " 'needs two overcome to challenges' -> 'needs to overcome two challenges' "
>
> A: Thank you. We will fix this typo.

---

> > ### Comment · Reviewer_et3m · 2021-08-22
> > **Response to authors rebuttal of paper 1341**
> >
> > I thank authors for their rebuttal. After reading all the reviews and rebuttals, I believe that the paper is above the acceptance threshold and hence it can be accepted. I have updated the rating post rebuttal.

---

> > > ### Author Response · Authors · 2021-08-22
> > > **Response to Reviewer (et3m)**
> > >
> > > Thank you very much.

---

### Decision · Program_Chairs · 2021-09-27

**Decision:**

Accept (Poster)

**Comment:**

This work proposes a new method for examining the patterns learned by intermediate layers of a DNN, by projecting features into a lower dimensional space that preserves discriminative structure. The reviewers were consistently positive about the paper’s novelty and impact, though some concerns were raised regarding clarity. The authors have done a good job of satisfying reviewer concerns in their rebuttals, and I trust that the authors can incorporate these clarifications into the final version without radically reshaping the paper. All in all, interesting and exciting work.